# Analysis of immune, microbiota and metabolome maturation in infants in a clinical trial of *Lactobacillus paracasei* CBA L74-fermented formula

Paola Roggero[1,2 ✉], Nadia Liotto[1], Chiara Pozzi [3], Daniele Braga [3], Jacopo Troisi [4,5,6], Camilla Menis[1,2], Maria Lorella Giannì [1,2], Roberto Berni Canani[7,8,9,10], Lorella Paparo[7,8], Rita Nocerino[7,8], Andrea Budelli[11], Fabio Mosca[1,2] & Maria Rescigno [3,12 ✉]

Mother's milk is the best choice for infants nutrition, however when it is not available or insufficient to satisfy the needs of the infant, formula is proposed as an effective substitute. Here, we report the results of a randomized controlled clinical trial (NCT03637894) designed to evaluate the effects of two different dietary regimens (standard formula and *Lactobacillus paracasei* CBA L74-fermented formula) versus breastfeeding (reference group) on immune defense mechanisms (primary endpoint: secretory IgA, antimicrobial peptides), the microbiota and its metabolome (secondary outcomes), in healthy full term infants according to the type of delivery ($n = 13$/group). We show that the fermented formula, safe and well tolerated, induces an increase in secretory IgA (but not in antimicrobial peptides) and reduces the diversity of the microbiota, similarly, but not as much as, breastmilk. Metabolome analysis allowed us to distinguish subjects based on their dietary regimen and mode of delivery. Together, these results suggest that a fermented formula favors the maturation of the immune system, microbiota and metabolome.

[1] Neonatal Intensive Care Unit, Fondazione IRCCS Ca' Granda Ospedale Maggiore Policlinico (IRCCS), Milan, Italy. [2] Department of Clinical Sciences and Community Health, University of Milan, Milan, Italy. [3] Humanitas Clinical and Research Center—IRCCS, Via Manzoni 56, 20089 Rozzano, Milan, Italy. [4] Theoreo Srl, Via degli Ulivi 3, 84090 Montecorvino Pugliano, SA, Italy. [5] European Biomedical Research Institute of Salerno (EBRIS), Via S. de Renzi, 3, 84125 Salerno, SA, Italy. [6] Department of Medicine, Surgery and Dentistry, "Scuola Medica Salernitana", Neuroscience Section, University of Salerno, Baronissi, SA, Italy. [7] Department of Translational Medical Science, University Federico II, Naples, Italy. [8] ImmunoNutritionLab at CEINGE-Biotecnologie Avanzate s.c.ar.l., University Federico II, Naples, Italy. [9] European Laboratory for the Investigation of Food-Induced Diseases, University Federico II, Naples, Italy. [10] Task Force on Microbiome Studies, University Federico II, Naples, Italy. [11] School of Engineering, Niccoló Cusano University, Rome, Italy. [12] Humanitas University Department of Biomedical Sciences, Via Rita Levi Montalcini 4, 20090 Pieve Emanuele, Milan, Italy. ✉email: paola.roggero@unimi.it; maria.rescigno@hunimed.eu

Exclusive breastfeeding is recommended from birth to 6 months as the normative standard for infants' nutrition (http://www.who.int/nutrition/topics/exclusive_breastfeeding/en/). Human milk provides protection against infections[1], promotes the development of intestinal microbiota, mainly *Bifidobacteria* that normally colonise the gut of breastfed infants[2] and exerts a crucial role in the development of the gut immune system[3]. When breast milk is not available or it is not sufficient to satisfy the nutritional needs, infant formula is proposed as an effective substitute dietary stategy[4].

Human milk substitutes should have nutritional and functional characteristics as close as possible to those of human milk. It is known that nutrition in early life plays an important role in the gastrointestinal tract immune and microbiota development and function[5]. The mode of delivery, feeding and antibiotic treatment, for instance, can all have an impact on microbiota composition early in life[6–8]. Hence, identification of formulas that better substitute for human milk, wherever the latter is unavailable, is imperative for the correct development of the newborn.

The mammary gland of non-human primates has been shown to harbour microbial DNA whose composition is regulated by the diet[9]. Similarly, in the mouse mammary gland, microbial DNA has been observed together with an immunoglobulin (Ig)A-enriched immune response[10]. The presence of microbial DNA does not necessarily indicate colonisation by an indigenous microbiota; however, the finding that breast milk and infant gut harbour microbial strains carrying antibiotic-resistance genes (ARGs) identical to those found in their own mother's gut microbiome, suggests that during pregnancy or lactation, gut microbes or their constituents translocate to the mammary gland[11,12]. Consistently, in non-human primates, the mammary gland harbours bile acid analogues and microbial bioactive compounds, which might be released in milk[9]. Further, the human milk also contains a highly diverse microbiota, which are transferred during lactation, and are fundamental for the establishment of the infant microbiota[13,14]. A study on 11 specific groups of the microbiota analysed by fluorescent in situ hybridisation (FISH) in the faeces of formula-fed versus breastfed infants has shown that formula-fed newborns have less-than-half abundance of *Bifidobacterium*[15] and a more diverse microbiota[15]. In line with this, a recent meta-analysis has shown that microbiota diversity increases in infants not fed exclusively with breast milk, suggesting that the introduction of infant formula may fail to control microbiota composition[16].

To improve the formula and to make it more functionally similar to breast milk, probiotic-enriched formulas have been proposed. The rationale was to provide bacteria (*Lactobacillus rhamnosus* GG, LGG) in the attempt to promote the establishment of a beneficial gut microbiota[17]. In this study, the presence of LGG fostered the growth of the newborns and the colonisation by *Lactobacilli*, when compared with standard formula, but this was not comparable with breast milk. As the microbiota are established in the first few months of life, it is not clear what could be the long-term effect of colonisation by a particular species of *Lactobacillus*. Indeed, we have described that secretory immunoglobulin A (sIgA) controls the diversification of the microbiota, and it is not clear whether driving the induction of sIgA towards one probiotic may interfere with the correct diversification of the whole microbiota[18]. Consistently, a report has recently shown that the use of probiotics after antibiotic treatment does not allow the correct mucosal microbiota reconstitution[19]. Supplementation of prebiotics to infant formula has also been proposed[20]. The human milk is rich in oligosaccharides[21,22], which shape the infant microbiome[23]. Thus, often infant formulas are supplemented with prebiotics, such as synthetic oligosaccharides like galacto-oligosaccharides, fructo-oligosaccharides and polydextrose[24] that have a different

structure from those of human milk, and cannot completely mimic the anti-inflammatory activity of human milk-derived oligosaccharides in vitro[25]. They favour the development of the gut microbiota, and in particular of *Bifidobacteria*, but what is the effect on the immune system has not been analysed in vivo[26]. Synthetic oligosaccharides structurally identical to those of human milk have recently become available on the market showing prebiotic and anti-infective properties when added to formulas[27].

A new microbiota-derived class of compounds is called postbiotics. Postbiotics are the metabolites released by bacteria during food fermentation[28,29]. The nature and mixture of postbiotics depend on the strain of fermenting bacteria, and on the matrix used for fermentation. Most of the beneficial activities of probiotics are mediated by postbiotics[29]. Hence, postbiotics may substitute for probiotics by providing directly the active components, which can mediate their effects on the immune system and the microbiota. A natural way to provide postbiotics is via food fermented with specific microbiota strains. We have previously studied the immunomodulatory properties of a formula fermented with an infant-derived gut microbiota strain of *Lactobacillus paracasei* (CBA L74) both in preclinical[30] and clinical studies in young children[31,32]. These studies have shown that the fermented formula stimulated the intestinal production of sIgA, and of innate immunity peptides, such as human alpha-defensins (HNP 1–3), human beta-defensin 2 (HBD-2) and cathelicidin LL-37, resulting in protection against infectious diseases[31,32].

However, there is no report that compares the activity of breast milk, formula or fermented formula rich in microbial metabolites on the defence mechanisms, metabolome and microbiota of the newborn. Here in a randomised clinical trial of newborns, we compare the activities of two different dietary regimens (standard formula and *Lactobacillus paracasei* CBA L74-fermented formula) with the reference group of breastfed infants on immune defence mechanisms (antimicrobial peptides, sIgA), the microbiota and its metabolome. Given the initial imprinting conferred to the microbiota by the type of delivery[6–8], in our cohort, we also consider the mode of delivery.

## Results

**Fermented and standard formula similarly support infant growth.** Seventy-eight full-term infants were included in a randomised clinical trial comparing two dietary regimens versus breastfeeding in naturally or caesarean section-delivered newborns (Supplementary Fig. 1, Supplementary Table 1). Newborns were randomised to receive until the third month of age a standard formula fermented with *L. paracasei* CBA L74 (Fermented formula F group) or a standard formula (Standard formula S group) in comparison with breast milk (reference group) (Fig. 1).

We analysed the anthropometric measurements of infants enrolled in the two study groups at each time point of the study (Table 1), and we did not find any statistically significant difference between the two study groups at each time point of the study. Infants fed with either formula S or formula F showed similar body composition at enrolment and at 3 months (visit 2).

Further, no differences were observed comparing the prevalence of gastrointestinal symptoms in the two formula study groups throughout the study, suggesting that both products were well tolerated (Table 2).

**Fermented formula promotes secretory IgA production.** To investigate the induction of an immune defence programme upon feeding with fermented or standard formula, we analysed the levels of sIgA and of innate immunity antimicrobial peptides,

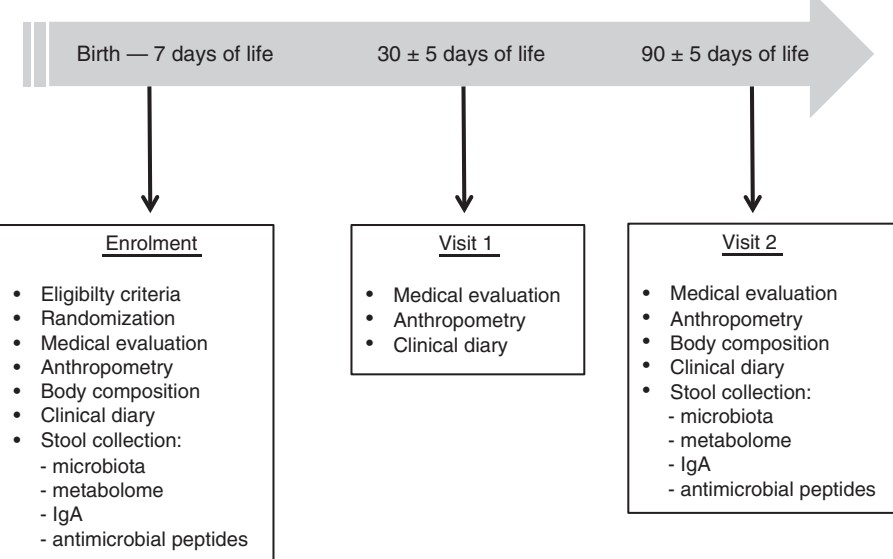

**Fig. 1 Study procedures.** Procedures related to collection of biological samples, medical evaluation and anthropometric measurements are shown in a timeline.

such as α-defensins (HNP 1–3), β-defensin 2 (HBD-2) and cathelicidin LL-37 in the faecal samples collected at enrolment and at visit 2. As shown in Fig. 2a, b, sIgA was present only in the reference group at enrolment. As enrolment was within 1 week from delivery (mean age was 3.3 ± 1.6 days), we can claim that the observed sIgA could be of maternal origin and in particular deriving from breast milk. Interestingly, formula F-fed infants, irrespective of the mode of delivery, reached values of sIgA similar to those of the reference group at visit 2. On the contrary, formula S-fed infants displayed little increase (Fig. 2) even when considering each infant separately (Supplementary Fig. 2). Overall, these data indicate that a fermented formula can drive sIgA production by the newborns.

As shown in Table 3, the levels of antimicrobial peptides, i.e., α-defensins (HNP 1–3), β-defensin 2 (HBD-2) and cathelicidin LL-37, were within the range observed in the reference group, without changes throughout the study, irrespective of the mode of delivery, indicating that antimicrobial peptides were not modulated during the study period.

**Fermented formula reduces microbial diversity at 3 months.** To study the effect of the two dietary regimens on the microbiota of infants, we analysed faecal samples at enrolment (T0) and at visit 2 (T2) and compared them with the reference group. We isolated bacterial DNA following a protocol established in the laboratory and performed microbiome analyses (16S rRNA analysis)[18]. Given the very young age of babies at enrolment, we could not collect the faeces from all of the newborns, and hence the number of samples was reduced to 54 (9 infants per group) to ensure a consistent number throughout the study groups. Chao1 alpha-diversity index allowed us to estimate the number of types of microorganisms found in each individual sample, and did not show statistically significant differences among groups between T0 and T2, except for vaginally delivered newborns fed the standard formula. Considering that the children in the reference group have been breastfed for a few days before enrolment, this may account for the difference already at T0 in this group. However, we observed a trend towards a reduced variability in breastfed infants at T2 as compared with T0, irrespective of the mode of delivery. By contrast, faeces from vaginally delivered

newborns fed the standard formula had a statistically significant increase in microbiota diversity at T2 (Fig. 3a).

Principal coordinates analysis (PCoA) (Fig. 3b) showed that at enrolment, the OTU of the different groups tended to separate from those of breastfed infants born by vaginal delivery (red circles), and clusterised into two groups depending on mode of delivery. This separation was lost at visit 2. As at T0 infants have already been breastfed for a few days, we cannot exclude that the clusterisation observed at enrolment by vaginally delivered breastfed infants may be attributed to both the microbiota associated with the mode of delivery and that contributed by the breast milk itself.

Analysis of variance between groups at genus level and Tukey's honestly significant difference (TukeyHSD) post hoc analysis displayed very few statistically significant differences at T0. Among these, *Bifidobacteria* were significantly more abundant in vaginally delivered as compared with caesarean-delivered infants independently of the feeding mode (one-way ANOVA, P = 0.0018). The presence of *Bifidobacteria* was null in most of the samples, except in vaginally delivered infants: breastfed (one null out of nine), formula F-fed infants (three null out of nine) and formula S-fed infants (four null out of nine), indicating that vaginal delivery and early breastfeeding favour the enrichment of *Bifidobacteria*. However, these differences were no longer significant at T2 where most of the infants displayed higher levels of *Bifidobacteria* even when analysed per each individual subject (Fig. 3c). This indicates that early colonisation by *Bifidobacteria* is mostly related to the type of delivery; over time, this group is similarly enriched independently of the mode of delivery or nutrition, even though breastfeeding favours the enrichment of *Bifidobacteria* early in life in vaginally delivered babies.

Consistent with an increased diversity in formula S-fed infants, when we analysed the effect of mode of delivery and feeding at visit 2, we observed several genera that characterised infants fed with formula S, particularly if vaginally delivered. This suggests that while vaginally delivered infants fed formula F had a more homogeneous microbiota, those fed the formula S had a more heterogeneous microbiota. We observed a statistically significant expansion of *Ruminococcus 2* and Erysipelotrichaceae in faeces of vaginally delivered formula S-fed infants, and a reduction of *Bacteroides* and *Parabacteroides* compared with vaginally delivered breastfed infants

**Table 1 Anthropometric measurements according to the type of feeding.**

| Parameters | Enrolment (0–7 DOL) | | | Visit 1 (30 ± 5 DOL) | | | Visit 2 (90 ± 5 DOL) | | |
|---|---|---|---|---|---|---|---|---|---|
| | Weight (g) | Length (cm) | Head circumference (cm) | Weight (g) | Length (cm) | Head circumference (cm) | Weight (g) | Length (cm) | Head circumference (cm) |
| Formula S (n = 26) | 3010 ± 303 | 49.3 ± 1.7 | 34.0 ± 1.0 | 4112 ± 433 | 52.9 ± 2.2 | 36.6 ± 1.0 | 5935 ± 667 | 59.9 ± 2.1 | 39.9 ± 1.2 |
| Formula F (n = 26) | 3041 ± 350 | 49.3 ± 1.5 | 34.2 ± 1.2 | 4160 ± 509 | 53.1 ± 1.8 | 37.5 ± 2.6 | 5885 ± 647 | 60.2 ± 2.0 | 40.2 ± 1.4 |
| Breast milk (n = 26) | 3105 ± 381 | 50.0 ± 1.7 | 34.1 ± 1.2 | 4225 ± 642 | 54.2 ± 2.1 | 37.5 ± 2.0 | 6082 ± 885 | 61.3 ± 2.6 | 40.6 ± 1.9 |

Analysis of variance was used to assess differences in continuous variables between groups.
No differences in anthropometric measurements were found among the two study groups (Formula S vs Formula F) at each study point.

**Table 2 Gastrointestinal tolerance.**

| | Spitting, n (%) | Vomiting, n (%) | Colic, n (%) | Daily stool frequency mean (SD) |
|---|---|---|---|---|
| *Visit 1* | | | | |
| Formula S | 4 (15.5) | 1 (3.8) | 12 (46) | 1.4 ± 0.7 |
| Formula F | 7 (27) | 3 (11.5) | 7 (27) | 1.4 ± 0.7 |
| Breast milk | 13 (52.0) | 2 (8.0) | 8 (32) | 3.4 ± 1.5 |
| *Visit 2* | | | | |
| Formula S | 5 (19.3) | 1 (3.8) | 5 (19.3) | 1.1 ± 0.3 |
| Formula F | 8 (31) | 0 (0) | 4 (15.4) | 1.4 ± 0.7 |
| Breast milk | 7 (28) | 1 (4) | 4 (16) | 2.2 ± 0.9 |

The $\chi^2$ test was performed to compare discrete variables among groups ($n = 26$/group). No differences in signs and symptoms of gastrointestinal intolerance were found among the two study groups (Formula S vs Formula F) during the study period.

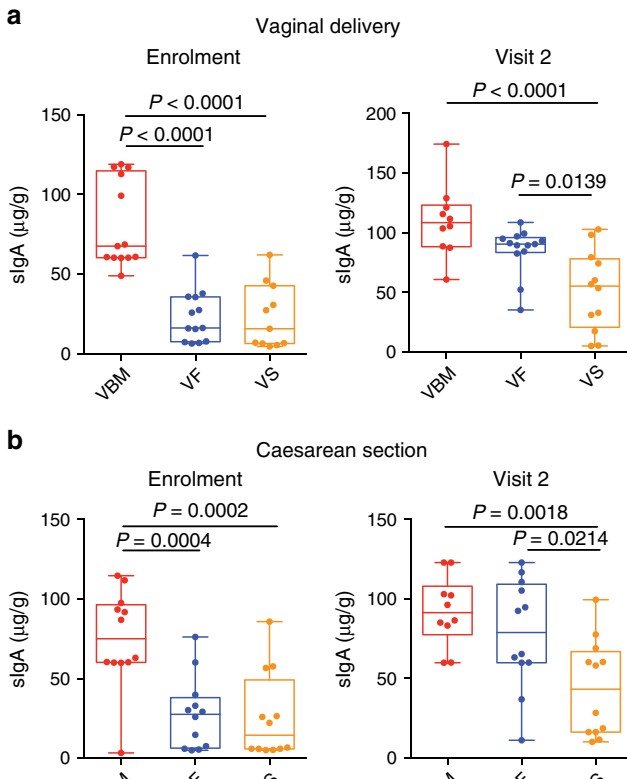

**Fig. 2 Fermented formula drives sIgA production similarly to breastfeeding. a**, **b** sIgA levels according to mode of delivery (vaginal in **a** and caesarean section in **b**) and type of feeding were determined in faecal samples. $n = 13$ (VBM and VF) and $n = 12$ (VS, CBM, CS and CF) biologically independent samples. Significance determined by one-way ANOVA using TukeyHSD post hoc test. Box plots show the interquartile range (IQR), the horizontal lines show the median values and the whiskers indicate the minimum-to-maximum range. CBM caesarean delivery breast milk feeding, CF caesarean delivery fermented formula feeding, CS caesarean delivery standard formula feeding, VBM vaginal-delivery breast milk feeding, VF vaginal-delivery fermented formula feeding, VS vaginal-delivery standard formula feeding. Source data are provided as a Source Data file.

(Supplementary Fig. 3, Supplementary Table 2). Moreover, *Clostridium innocuum* was enriched in formula S-fed infants, irrespective of the mode of delivery, while *Veillonella* was reduced in formula-fed as compared with breastfed infants's microbiota, irrespective of the mode of delivery (Supplementary Table 3).

**Table 3 Antimicrobial peptides at each study point according to randomisation group and mode of delivery.**

|  | α-defensins (ng/g) | | β-defensin2 (ng/g) | | Cathelecidin LL-37 (ng/g) | |
|---|---|---|---|---|---|---|
|  | T0 | T2 | T0 | T2 | T0 | T2 |
| *Vaginal delivery* | | | | | | |
| Formula S (median) | 1.65 ± 2.4 | 0.29 ± 1.7 | 68.2 ± 29.8 | 70.3 ± 34.7 | 1.8 ± 2.5 | 1.7 ± 4.0 |
| Formula F (median) | 2.2 ± 2.2 | 3.0 ± 2.8 | 69.7 ± 31.1 | 96.9 ± 82.6 | 1.9 ± 0.9 | 2.6 ± 3.3 |
| Breast milk (range) | 0.13–9.55 | 0.24–5.36 | 23.2–116.5 | 34.8–123.1 | 1.0–4.4 | 1.7–28.4 |
| *Caesarean section* | | | | | | |
| Formula S (median) | 1.3 ± 1.2 | 0.2 ± 0.4 | 55.6 ± 25.1 | 65.6 ± 21.4 | 1.8 ± 1.3 | 1.4 ± 1.6 |
| Formula F (median) | 1.6 ± 1.4 | 0.1 ± 2.4 | 62.8 ± 22.9 | 64.7 ± 23.3 | 1.6 ± 1.9 | 3.7 ± 1.8 |
| Breast milk (range) | 0.5–4.9 | 0.04–10.6 | 11.3–119.1 | 3.78–108.7 | 1.2–11.9 | 1.6–14.6 |

All values are expressed as median ± SD or range. A repeated measures Friedman non-parametric test was used to compare immunological parameters between groups ($n = 13$/group).

To exclude any possible counfounding element derived from differences at enrolment linked to the initial feeding with standard diet versus breast milk, we performed a statistical analysis that accounted for the individual baseline of each subject using a linear mixed model. Individual trends of the taxa abundances at T0 and T2 are shown in Supplementary Fig. 4. This analysis allowed us to pinpoint more taxa that were differentially represented in the different groups than when the microbiome was analysed in the whole-population groups. Some taxa, such as *Eggerthella* and Lachnospiraceae, were similarly regulated in both formula-fed caesarean- delivered infants and differed from breastfed infants. *Ruminococcus 2*, Erysipelotricha-ceae, bacteria belonging to Alphaproteobacteria class (gut metagenome and Clade I), and partly also Ruminococcaceae UCG 014, were enriched in vaginally delivered standard formula-fed infants at T2. This supports the difference in alpha-diversity observed in vaginally delivered babies fed a standard diet.

Overall, the microbiota analysis shows that formula-fed infants are distinguishable from the reference group both at enrolment and at visit 2. However, over time, while the microbiota tend to become more homogeneous in formula F-fed infants, the microbiota of vaginally delivered infants fed the formula S are characterised by a higher diversity that is particularly evident and statistically significant at genus level.

**The metabolic profile groups infants according to diet**. At enrolment, we found that formula-fed infants demonstrated a differentiation in faecal metabolites compared with breastfed infants. In particular, the metabolome of breastfed infants was characterised not only by monosaccharides, such as sorbose and rhamnose, but also by fatty acids such as dodecanoic acid (Fig. 4a). This is likely due to metabolites contained within the breast milk that are transferred to the newborns. At visit 2, although faecal metabolites of formula-fed infants were still different compared with those of breastfed infants, the metabolome of formula F-fed infants was closer to that of the reference group. Specifically, in faecal samples of formula S-fed infants, we observed a higher amount of carboxylic acids, such as fumaric and acetic acid, and saturated fatty acids such as myristic acid, than in breastfed and formula F-fed infants (Fig. 4b). A list of metabolites statistically different among groups, as calculated by ANOVA, is shown in Table 4 (T0) and Table 5 (T2).

We then analysed the metabolome taking into account both the mode of delivery and the type of feeding. As shown in Fig. 4c, the mode of delivery had an important role in modulating the metabolites' differentiation. The reference group born from a vaginal delivery was clearly separated and distant from the other groups with a different clusterisation of faecal metabolites already at enrolment, suggesting that both the mode of delivery and

feeding had an impact on metabolite composition already in the first week of life.

As shown in Fig. 4d, at visit 2, the reference group born by caesarean section was not able to achieve the metabolomic assessment of the reference group born by vaginal delivery. Importantly, although formula-fed infants remained different from breastfed infants, vaginally delivered formula F-fed infants had a pattern of faecal metabolites closer to that of breastfed infants compared with formula S-fed infants. The same was true when analysing caesarean section-delivered infants.

We then analysed metabolites showing different concentration trends (T0–T2) according to delivery–diet groups accounting for fixed effects of diet and delivery and random effects of subjects, thus taking into account the individual baseline of each subject. In Supplementary Fig. 5, we reported 20 statistically significant different metabolites (by ANOVA among the diet-delivery groups, at least one group different from the others, $P < 0.05$) for each individual infant over the course of time (T0–T2). Even taking into account baseline levels for individual subjects, metabolites in formula S-fed infants clearly demonstrated a trend that was different from that of the reference group, independent of the type of delivery. By contrast, trends of metabolites associated with the formula F-fed group had values that were in some cases more similar to those of formula S and in others more similar to the reference group. In particular, amino acids, such as valine and threonine, followed a trend similar in caesarean-delivered formula S- and formula F-fed infants, but in the latter group, they were less abundant and significantly different ($P < 0.05$). Similarly, 2-keitoisocaproic acid, threose, 5-hydroxyindoleacetic acid and serine also followed a trend similar in formula S- and in formula F-fed infants. By contrast, metabolites, such as glyceril glycoside, glyceric acid, galactose and arabinose, displayed trends in caesarean-delivered formula F-fed infants more similar to those of the reference groups. We also found some metabolites that were reduced or absent in formula-fed infants as compared with the reference group, such as aspartic acid, methionine, arabitol, mannobiose, 2–3-dihydroxy-2-methyl-propanoic acid, propylene glycol, pentanedioic acid, stearic acid, galactofuranose and ribose. These metabolites were found significant primarily in the vaginal-delivery reference group, confirming their separation as a group versus all the other groups (Supplementary Fig. 5).

Overall, these results indicate that the metabolome can clearly distinguish newborns, both for their dietary regimen and for mode of delivery, and is a more sensitive way than the microbiome to stratify individuals according to their dietary regimen, at least when simple diets are considered.

**Correlation between metabolites and genera of the microbiota**. We then analysed whether the differences observed in the diet

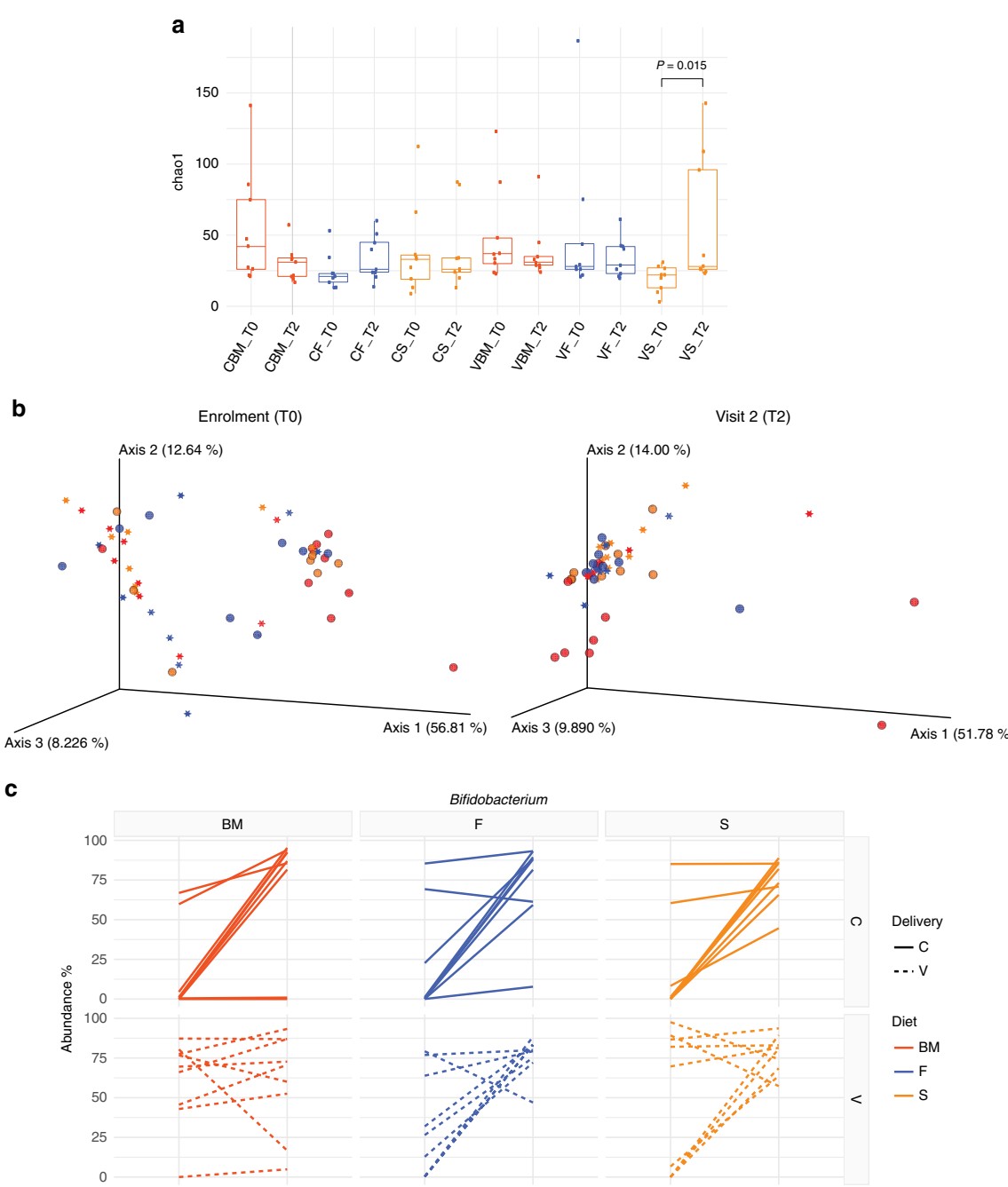

**Fig. 3 Fermented formula but not standard formula reduces microbiota diversity over time. a** Chao1 index of bacterial α-diversity of the faecal microbiota of infants at enrolment (T0) and visit 2 (T2). Significance determined by Wilcoxon signed-rank test for repeated measurements (two-tailed). Box plots show the interquartile range (IQR), the horizontal lines show the median values and the whiskers extend from the hinge no further than 1.5*IQR. CBM: caesarean delivery breast milk feeding, CF caesarean delivery fermented formula feeding, CS caesarean delivery standard formula feeding, VBM vaginal-delivery breast milk feeding, VF vaginal-delivery fermented formula feeding, VS vaginal-delivery standard formula feeding. **b** PCoA analysis of weighted UniFrac distance (β-diversity) of gut microbial communities according to the mode of delivery (caesarean—star; vaginal—sphere) and the type of feeding (BM—red; F—blue; S—orange). Left, enrolment; right, visit 2. **c** Interaction plots of *Bifidobacterium* abundance according to diets and delivery mode at T0 and T2. Each line represents individual newborns (solid lines caesarean delivery, C; dotted lines vaginal delivery, V). Significance determined accounting for individual baselines with a linear mixed model and ANOVA to test the interaction between groups and time, *P* = 0.011. BM breast milk feeding, F fermented formula feeding, S standard formula feeding. Source data of panels **a** and **c** are provided as a Source Data file.

groups were ascribable to correlations between microbial genera and metabolites. We limited this analysis to T2 and to the metabolites found to be statistically significantly different in the population as a whole (by ANOVA, Table 5 or Partial Least Squares Discriminant Analysis—PLS-DA, Fig. 4b). As shown in Fig. 5, the first striking observation is that we detected four

different clusters. One (box 1) included metabolites rich in the faeces of the reference group and sometimes of the fermented formula-fed group. These metabolites correlated with a set of genera, including Clostridiaceae, *Subdoligranulum*, *Butyricicoccus*, Ruminococcaceae UCG 014, *Faecalibacterium*, *Eubacterium coprostanoligenes*, *Bacillus*, *Escherichia–Shigella*,

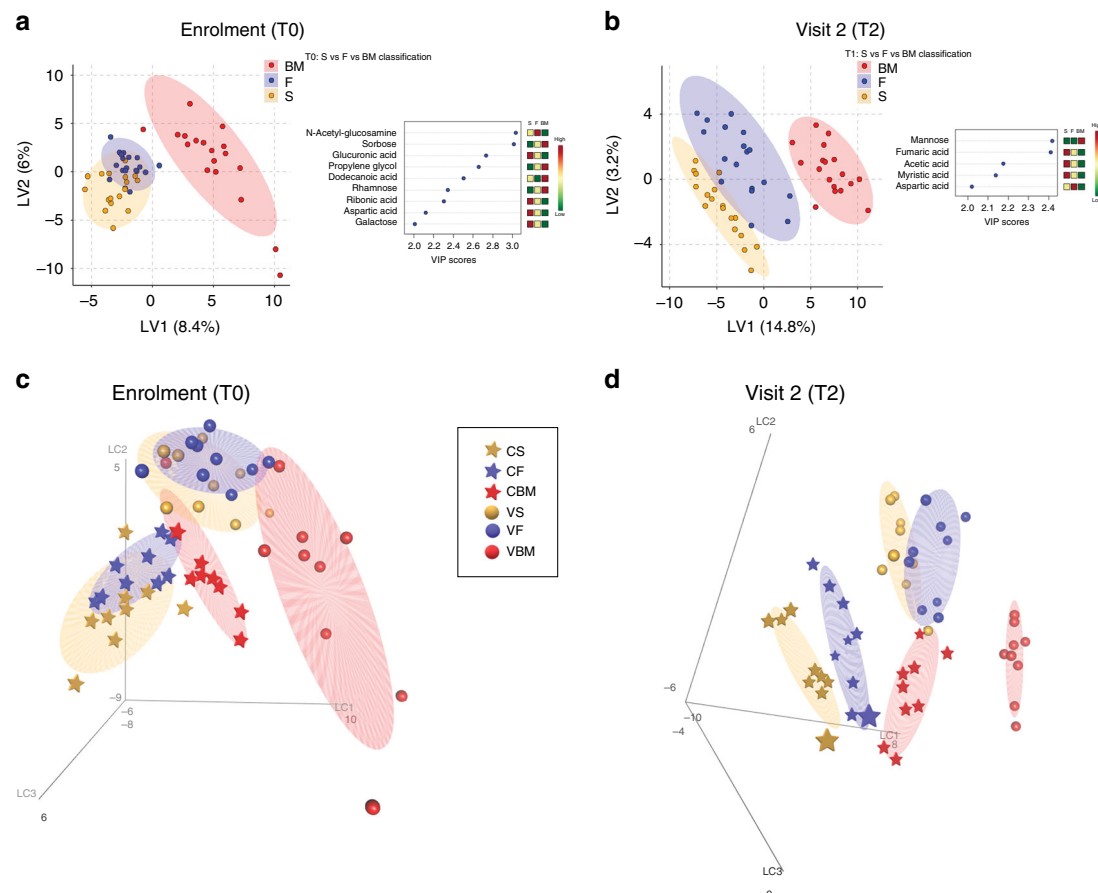

**Fig. 4 Metabolome clearly distinguishes newborns based on their dietary regimen. a, b** Metabolomic assessment at enrolment (**a**) and at visit 2 (**b**) according to the type of feeding. Left, a PLS-DA analysis is shown separating infants breastfed (BM, red dots), fed formula F (F, blue dots), or formula S (S, orange dots). On the right are shown those metabolites that differentiate the three diets (VIP score > 2) and their mean relative concentration. **c, d** Metabolomic assessment (PLS-DA 3D) at enrolment (**c**) and at visit 2 (**d**) according to the type of feeding (BM—red; F—blue; S—orange) and mode of delivery (caesarean—star; vaginal—sphere). CBM caesarean delivery breast milk feeding, CF caesarean delivery fermented formula feeding, CS caesarean delivery standard formula feeding, VBM vaginal-delivery breast milk feeding, VF vaginal-delivery fermented formula feeding, VS vaginal-delivery standard formula feeding.

**Table 4 List of significant metabolites among groups at T0.**

|  | *F* value | *P* value | −LOG10(p) | FDR | Fisher's LSD[a] |
|---|---|---|---|---|---|
| Galactitol-2-amino-2-deoxy-N-acetyl | 5.1919 | 0.00068992 | 3.1612 | 0.034874 | VBM-CF; VBM-CBM; VBM-VS; VBM-VF |
| N-Acetyl-ᴅ-glucosamine | 5.0968 | 0.00079258 | 3.101 | 0.034874 | VBM-CF; VBM-CBM; VBM-VS; VBM-VF |

ANOVA analysis (FDR < 0.05) at T0 according to diet and delivery groups.
[a]Comparisons between groups with significant post hoc analysis (Fisher's LSD < 0.05).

*Flavonifractor*, *Microbacterium*, Ruminococcaceae UCG 013, Ruminococcaceae, Coriobacteriales incertae and Pasteurellaceae. Interestingly, none of these bacteria individually associated with breastfed infants' microbiota; however, given the overlap of metabolites associated with these bacteria, it is likely that they may contribute as a group. The other two groups included metabolites enriched in the faecal metabolome of formula S-fed infants: threonin, fumaric acid, 2-hydroxyisocaproic acid, leucine, valine (box 3) and threose 2, 5-dimethylaminomethyl-1H-pyr-role-2-carboxylic acid–methyl ester and myristic acid (box 4). These groups of metabolites correlated with two different groups of genera, including either Muribaculaceae spp., *Faecalibaculum*, Erysipelotrichaceae and *Ruminococcus 2* (box 3), or *Blautia*, *Eggerthella*, Lachnospiraceae, *Clostridium innocuum* and *Var-ibaculum* (box 4). Notably, *Clostrodium innocuum*,

Erysipelotrichaceae, *Ruminococcus 2*, *Eggerthella* and Lachnos-piraceae were enriched in formula-fed infants. We also observed a cluster of genera (box 2) that were inversely correlated with some metabolites typically found in the faeces of formula S-fed infants (threonin, fumaric acid, 2-hydroxyisocaproic acid, leucine and valine). These genera included *Parabacteroides*, *Veillonella*, *Hae-mophilus* and *Bacteroides*. There were no specific metabolites characterising only the formula F, but some were more similar to breastfed and others to formula S-fed infants, confirming their mixed nature in-between the two diets.

## Discussion

Identifying a good substitute for breast milk is imperative to offer newborns, that cannot be breastfed, a nutrition regimen that can

**Table 5 List of significant metabolites among groups at T2.**

| | F value | P value | −LOG10(p) | FDR | Fisher's LSD[a] |
|---|---|---|---|---|---|
| L-Ascorbic acid | 8.3458 | 9.77E-06 | 5.0101 | 0.00087 | CF-CS; VS-CS; VF-CS; CF-CBM; CF-VBM; VS-CBM; VF-CBM; VS-VBM; VF-VBM |
| N-acetyl-D-glucosamine | 7.6955 | 2.23E-05 | 4.6516 | 0.000992 | VBM-CS; VBM-CF; VBM-CBM; VBM-VS; VBM-VF |
| L-Methionine | 6.9005 | 6.34E-05 | 4.198 | 0.001881 | CBM-CS; CBM-CF; CBM-VS; CBM-VF; CBM-VBM |
| Arabitol 4 | 6.5741 | 9.85E-05 | 4,0065 | 0.002192 | CBM-CS; VBM-CS; CBM-CF; VBM-CF; CBM-VS; VBM-VS; VBM-VF |
| L-Threonine | 6.2239 | 0.000159 | 3.7977 | 0.002836 | CS-CBM; CS-VF; CS-VBM; CF-VBM; VS-CBM; VS-VF; VS-VBM |
| 2-3-Butanediol | 5.9549 | 0.000232 | 3.6348 | 0.003439 | CBM-CS; CBM-CF; CBM-VS; CBM-VF; CBM-VBM |
| 5-Dimethylaminomethyl 1H-pyrrole-2-carboxylic acid–methyl ester | 5.1082 | 0.000779 | 3.1082 | 0.009822 | CS-CF; CS-CBM; CS-VS; CS-VF; CS-VBM |
| Galactitol-2-amino-2-deoxy- N-acetyl | 5.0231 | 0.000883 | 3.0541 | 0.009822 | CBM-CS; VBM-CS; CBM-CF; CBM-VS; CBM-VF; VBM-VS; VBM-VF |
| 2-Alpha-mannobiose 2 | 4.5294 | 0.001837 | 2.736 | 0.018161 | CS-CF; CS-CBM; CS-VS; CS-VF; VBM-CF |
| 2-3-Dihydroxy-2-methylpropanoic acid | 4.4544 | 0.002056 | 2.687 | 0.018296 | CBM-CS; VBM-CS; CBM-CF; VBM-CF; CBM-VS; CBM-VF; VBM-VS; VBM-VF |
| D-Allose 1 | 4.1187 | 0.00342 | 2.466 | 0.027669 | VBM-CS; VBM-CF; CBM-VF; VBM-VS; VBM-VF |
| L-Valine | 3.9443 | 0.004467 | 2.35 | 0.033133 | CS-CBM; CS-VBM; VS-CBM; VS-VBM |
| 2-Butenedioic acid | 3.8178 | 0.005429 | 2.2653 | 0.035468 | CS-CBM; CS-VF; CS-VBM; VS-CBM; VS-VBM |
| D-Psicofuranose | 3.7758 | 0.005793 | 2.2371 | 0.035468 | VBM-CS; CBM-CF; VBM-CF; CBM-VS; CBM-VF; VBM-VS; VBM-VF |
| 2-Hydroxyisocaproic acid | 3.7555 | 0.005978 | 2.2235 | 0.035468 | CS-CBM; CS-VF; CS-VBM; CF-CBM; CF-VF; CF-VBM |
| Arabinose | 3.6773 | 0.006749 | 2.1708 | 0.037539 | VS-CS; VS-CF; VS-CBM; VS-VF; VS-VBM |
| N-acetyl-D-galactosamine 3 | 3.596 | 0.007658 | 2.1159 | 0.040093 | VBM-CS; VBM-CF; VBM-CBM; VBM-VS; VBM-VF |
| Glyceryl-glycoside | 3.5171 | 0.008661 | 2.0624 | 0.042822 | VBM-CS; VBM-CF; VBM-VS; VBM-VF |
| Beta-L-Fucopyranose 2 | 3.4277 | 0.00996 | 2.0017 | 0.043722 | VBM-CF; VBM-CBM; VBM-VS; VBM-VF |
| 2-Ketoisocaproic acid | 3.421 | 0.010065 | 1.9972 | 0.043722 | CS-CBM; CS-VBM |
| D-Glucose 3 | 3.4052 | 0.010316 | 1.9865 | 0.043722 | VBM-CS; VBM-CF; VBM-CBM; VBM-VS; VBM-VF |
| L-Leucine | 3.3595 | 0.011084 | 1.9553 | 0.04484 | VS-CBM; VS-VBM |
| L-Threose 2 | 3.2792 | 0.012573 | 1.9006 | 0.048651 | CS-CBM; CS-VS; CS-VF; CS-VBM |

ANOVA analysis (FDR < 0.05) at T2 according to diet and delivery groups.
[a]Comparisons between groups with significant post hoc analysis (Fisher's LSD < 0.05).

foster the development of the immune system and the microbiota. In this study, we demonstrate that a formula fermented with *Lactobacillus paracasei* CBA L74 is safe and well tolerated. In addition, the results obtained in this RCT show that formula F-fed infants approached the acquired immunity (sIgA), gut microbiota and faecal metabolomic profiles of breastfed infants. sIgA is associated not only with acquired immunity but also with regulation of intestinal homoeostasis and microbiota[33]; thus, the fermented formula could improve intestinal homoeostasis and decrease the risk of developing infections. This is in line with the results of two recent clinical trials in young children exposed to a fermented formula that were more protected against infectious diseases[31,32].

Unlike sIgA, our results did not show any significant difference between groups in terms of innate immunity peptides (HNP 1–3, HBD-2 and LL-37). All of the values were within the range observed in the reference group, both at the time of enrolment and at visit 2. This is probably due to the enrolment of full-term healthy infants that did not report any infection during the time of the study. Indeed, antimicrobial peptides are increased after intestinal inflammation to carry out a microbicidal and chemotactic activity[34,35]. Our previous finding that preschool children fed with fermented formula displayed increased faecal antimicrobial peptides may be due to the capacity of the fermented product to potentiate the antimicrobial response, which is evident only in older children because they are exposed to pathogens during their daily community life[31,32].

The effect of the mode of delivery and, above all, the duration of this effect on modulating the gut microbiota is controversial[36]. For example, in a longitudinal study conducted in a cohort of 81 pregnant women enrolled at the third trimester and followed prospectively with their infants through 6 weeks after delivery, it was found that the infant's microbiota structure and function had expanded and diversified without differences between infants delivered vaginally or by caesarean section[36]. Consistently, we found that at enrolment, infants born by vaginal delivery had a significantly higher abundance of *Bifidobacteria*, which was even more evident in the reference group, whereas these differences were no longer significant at visit 2. Microbiota PCoA analysis did not reveal major clusterisation patterns among the groups, except for a trend to separation among infants by vaginal and caesarean delivery, and in breastfed infants born by natural delivery at enrolment. However, this pattern could not be observed in breastfed caesarean-delivered infants, suggesting that this difference may be attributed to both mode of delivery and feeding. We observed that standard formula-fed infants born by vaginal delivery had a more diversified microbiome compared with babies fed by fermented formula and the reference group. This confirms and extends an analysis performed on 11 specific groups of the microbiota analysed by fluorescent in situ hybridisation (FISH) in the faeces of formula-fed versus breastfed infants[15], and a meta-analysis comparing exclusively mixed-versus breastfeeding infants[16].

This more diversified microbiome was observed only at T2 and not at enrolment, even when taking into account subject-related baseline differences, indicating that it is likely the result of the mode of feeding. We observed in particular an enrichment in *Ruminococcus 2*, Erysipelotrichaceae, bacteria belonging to Alphaproteobacteria class (gut metagenome and Clade I) and partly also Ruminococcaceae UCG 014.

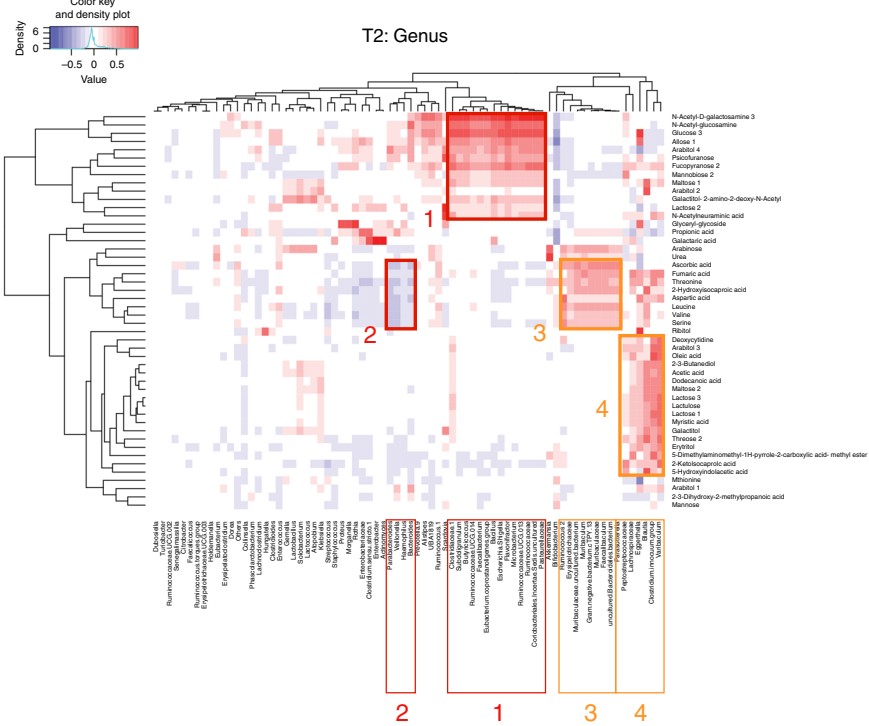

**Fig. 5 Microbe–metabolite correlation heatmap showing associations between bacterial genus and the ANOVA and VIP-score-selected metabolites.** Spearman correlation coefficients were calculated for pairwise combinations of microbial abundances at the genus level (from 16S rRNA gene sequencing data) and metabolite intensities. In red boxes are reported clusters of metabolites either enriched (box 1) or reduced (box 2) in vaginally delivered breastfed infants. Box 2 metabolites are associated with genera highly abundant in breastfed infants; in orange boxes (3 and 4) are highlighted clusters of metabolites that are enriched in formula S-fed infants. The corresponding genera correlating with these metabolites are also highlighted with the same colour and numbering code.

On the other hand, the metabolomic analysis clearly showed that formula-fed infants displayed a differentiation in faecal metabolites compared with breastfed infants at enrolment. As mentioned above for microbiota analysis, it is likely that since the enrolment occurs within 7 days from birth, this breastfeeding period is sufficient to already shape the microbiota and, consequently, also their metabolome. Alternatively, and not mutually exclusive, it may also reflect the metabolome of breast milk. Indeed, the metabolome of breastfed infants was rich not only in monosaccharides, such as sorbose and rhamnose, but also fatty acids such as dodecanoic acid, which are likely to be of maternal milk origin rather than derived from the microbiota. Indeed, this difference was maintained at 3 months.

In accordance with previous studies[37], our study confirms that the mode of delivery influences metabolomics. Specifically, we found that breastfed infants born by vaginal delivery had a metabolomic assessment throughout the study, which was different even from that of breastfed infants born by caesarean section. Despite these findings, it seems that the fermented formula could modulate the metabolomic pattern, partially balancing the role of mode of delivery, and making it more similar to that of the reference group of breastfed infants. For example, we found that formula F-fed infants had levels of myristic acid similar to those of breastfed infants, but reduced compared with formula S-fed infants. Myristic acid has been associated with the appetitive response in newborns[38]. However, we did not observe any change in body growth in the three different groups, suggesting that its presence does not seem to influence food intake, at least in this time frame. Interestingly, intraperitoneal injection of myristic acid in a concentration similar to that found in the amniotic fluid to rats has been shown to induce an anxiolytic-like

behaviour[39]. Reduction of myristic acid in the fermented formula may give a health benefit to the newborn.

We also found groups of metabolites that correlated with a particular set of bacterial genera. In particular, we observed a series of metabolites enriched in formula S-fed infants that were correlating with *Clostridium inocuum group*, Erysipelotrichaceae, *Eggerthella*, Lachnospiraceae and *Ruminococcus 2*, which were enriched in formula-fed infants. Unfortunately, little is known on most of these bacteria, and it is hard to infer possible advantages or disadvantages of having an enrichment of these bacteria. *Ruminococci*, for instance, have been shown to produce short-chain fatty acids, such as propionate and butyrate[40], that have several beneficial effects, both on epithelial cell proliferation and barrier functions[41]; however, there are no specific reports on *Ruminococcus 2*. Similarly, regarding *Clostrodium innocuum*, there is only one report showing that it is associated with colo-bronchial fistula formation in Crohn's disease together with other potential pathogens[42]. Erysipelotrichaceae have been associated not only with inflammatory states of the gut[43], but also with tumour protection[44], but again individual species might have different activities. More in-depth analysis should be carried out to deepen our knowledge on the enrichment of these genera. The reference group was characterised by the presence of *Bacteroides*, *Parabacteroides* and *Veillonella* that inversely correlated with a set of standard-formula-enriched metabolites (threonin, fumaric acid, aspartic acid, 2-hydroxyisocaproic acid, leucine and valine). 2-Hydroxyisocaproic acid is a metabolic intermediate of L-leucine, while leucine and valine are branched amino acids, and are the bases for the production of other essential amino acids, such as glutamate, glutamine and aspartate[45]. It is not clear why in the formula-fed group there should be increase in these amino acids

as they are highly represented in milk proteins[45]. One possibility is that *Bacteroides*, *Parabacteroides* or *Veillonella* might use up these amino acids or transform them, thus reducing their concentration, and hence since these taxa are not enriched in formula-fed infants, these amino acids accumulate. Notably, *Bacteroides* make up a substantial portion of the human gut microbiota, and are considered necessary to refine the gut environment and transforming it into one more hospitable for themselves and the other microrganisms[46]. They reduce, for instance, oxygen levels intracellularly reducing at the same time both inflammatory reactive oxygen species and generating a strictly anaerobic environment that favours the growth of anaerobic microbes[47]. In addition, *Bacteroides* are important for the digestion of complex molecules, including plant- or host-derived polysaccharides[46]. As mentioned above, the human breast milk is rich in complex carbohydrates that are indigestible by our enzymes, and this may explain the abundance of *Bacteroides* in the reference group[21,22]. *Veillonella*, on the other hand, uses lactate as its only source of carbon[48], and this may explain why lactose and lactulose are not abundant in the reference group.

We could not find increased abundance of genera correlating with metabolites characteristic of the reference group of breastfed infants. This could be due to a limitation of the study as we analysed too few individuals in each group, or to the observation that many genera contributed to the same sets of metabolites, and thus they could contribute as a whole to increase these metabolites without being individually dominant or enriched.

We confirmed literature data showing an impact of birth on microbiota composition[6–8], but we could not really distinguish the microbiota associated with the three groups, except for a few genera that were enriched in particular dietary regimens. Nevertheless, we could show that the diet had an impact on the metabolome, even with only 9 infants per group, and that there was homogeneity among the groups when taking into account the baseline for individual subjects. Another limitation of this study stands in the analysis of 16S rRNA rather than shotgun metagenomic sequencing, and this may have an impact on the identification of the microbiome at species level.

With all the limitations of this study, in spite of the short follow-up and small sample size, we observed significant differences in the three dietary regimens in terms of both the metabolome and sIgA induction. We also identified metabolites that positively or negatively correlated with bacterial genera, some of which were enriched in the corresponding diet. We found metabolites, such as rhamnose, sorbose and propylene glycol, that were found in large amounts in the faeces of the reference group, intermediate amounts in fermented formula-fed infants and low in standard formula-fed infants, but that did not correlate with any of the analysed microbial genera that were enriched in the different dietary regimens. This could be due to the involvement of bacterial genera not highly represented, or statistically modified, or to microbes of a different nature, such as eukarya that were not analysed in this report. Our results further confirm the incredible nutritional value of breast milk, and suggest that most of its beneficial activities may be provided by its microbiota-associated metabolites, and indicate that the combined analysis of the metabolome and microbiome may be more accurate than the microbiota alone in small-size groups to assess the effect of nutrition.

## Methods

**Study design**. A monocentric, randomised, double-blind, placebo-controlled, parallel group trial with reference group was performed between September 2015 and April 2016 in the Department of Neonatology of Fondazione IRCCS Ca' Granda, Ospedale Maggiore Policlinico of Milan. The study was approved by the Ethics Committee and conducted in accordance with Good Clinical Practice and the principles and rules of Declaration of Helsinki. Parents or legal caregivers provided written informed consent before the enrolment of their infants in the study. The study was prospectively registered in the Ethical Committee repository of Fondazione IRCCS Ca' Granda Ospedale Maggiore Policlinico [https://www.albopretorionline.it/irccs/alboente.aspx?ente=pol82365&piva=04724150968&sez=SCAD&anno=2016&categoria=Determinazioni]. This database is public, contains the essential information required by ICMJE, but is not electronically searchable. The study protocol document is available as Supplementary Data 1.

The trial was registered retrospectively in the Clinical Trials Protocol Registration System (ClinicalTrials.gov) with the identifier NCT03637894.

**Study subjects**. Healthy full-term infants were screened for participation in the study. Inclusion criteria were singleton, full-term infants (gestational age from 37/0 to 41/6 weeks), with a birth weight adequate for gestational age (>10th percentile and <90th percentile for gestational age) according to the World Health Organization growth charts (available at http://www.who.int/childgrowth/standards/en/), when entering the study. Exclusion criteria were the presence of congenital diseases, chromosomal abnormalities and/or conditions that could interfere with growth, such as brain, metabolic, cardiac and gastrointestinal diseases, perinatal infections, being born to a mother affected by endocrine and/or metabolic diseases or having a family history of allergic disease. The mean weight and gestational age at birth of infants enrolled were 3243 ± 372 g and 38.5 ± 1.0 weeks. The age range at enrolment was 0–7 days (3.3 ± 1.6 days). Therefore, as the mean weight and gestational age at birth were similar among groups, we did not perform any correction for further analyses.

**Intervention**. All the mothers of infants evaluated for the study were encouraged to breastfed their infants for at least 4 months: if they could not or intended not to breastfeed their infants, the study investigators asked them for their consent to participate in the study within 7 days after birth. Newborns were randomised to receive until the third month of age-standard formula containing 2.3 g/100 g of cow's milk powder fermented with the probiotic *L. paracasei* CBA L74 (formula F group) or standard formula (formula S group), which is equivalent to 0.3% in the ready to use infant formula. The reference group was constituted only by breastfed infants and not by mother's milk bottle-fed infants.

The composition of the study dietary products has already been described in ref. [31]. The powder was provided by Heinz Italia SpA, Latina, Italy, an affiliate of H. J. Heinz Company, Pittsburgh, PA, USA. The fermented milk was prepared from skimmed milk fermented with *L. paracasei* CBA L74, which was isolated from the faeces of healthy infants. The fermentation was initiated with $10^6$ bacteria, and stopped when reaching $5.9 \times 10^9$ colony-forming units/g (after a 15-h incubation at 37 °C). Live bacteria were then inactivated by heating at 85 °C for 20 s, and the formula was spray-dried. Study products were provided already prepared in tins containing 400 g of powder, and the packaging was similar between S and F formulas. Study products were stored at room temperature and in a dry environment.

The randomisation schedule was computer-generated and stratified on type of delivery using 2 computer-generated randomisation lists, 1 for vaginal-delivered infants and 1 for caesarean section-delivered infants. Study formulas were formulated into powder and were reconstituted at 13.3%, and the packages were the same and identified with letters to make them unrecognisable. They had similar energy and macronutrient contents, but they differed by the presence of fermented product. Specifically, the formula F content was energy 69 kcal/100 ml, fats: 3.6 g/100 ml, proteins: 1.4 g/100 ml and carbohydrates: 7.4 g/100 ml; formula S content was energy 68 kcal/100 ml, fats: 3.6 g/100 ml, proteins: 1.4 g/100 ml and carbohydrates: 7.3 g/100 ml. Infants fed with exclusive breast milk for the first 3 months of life represented the reference group.

**Procedures**. The study included three medical examinations: at enrolment (up to 7 days of life), at 30 ± 5 days of life (visit 1) and at 90 ± 5 days of life (visit 2). The procedures conducted at each study point are shown in Fig. 1. Biological samples were collected only at enrolment and visit 2, and were used for all the analyses.

**Growth parameters**. Anthropometric and body composition measurements were performed by the same medical investigator who was blinded to the allocated treatment. Body weight, length and head circumference were measured at birth, 1 and 3 months according to standard procedures[49,50]. Weight was measured on an electronic scale accurate to 0.1 g (PEA POD Infant Body Composition System, Cosmed, Italy). Body length was measured to the nearest 1 mm on a Harpenden neonatometer (Holtain, Crymych, UK). Head circumference was measured to the nearest 1 mm using non-stretch measuring tape. Body composition was assessed at enrolment and at 3 months using an air-displacement plethysmography (PEA POD Infant Body Composition System, COSMED, Italy). A detailed description of the PEA POD's physical design, operating principles, validation and measurement procedures is provided elsewhere[51–53].

**Diary**. All parents were instructed to fill a clinical diary. Specifically, they had to record any use of drugs, possible hospitalisations, possible adverse events, consumption of the study products and data regarding gastrointestinal tolerance. The

following indicators of tolerability were collected through multiple-choice questions: volume of formula intake, daily frequency of stool passage, episodes and type of vomit or spitting and episodes of flatulence and abdominal pain (infantile colics: defined as intermittent attacks of abdominal pain when the baby screams and draws up his/her legs, but is well between episodes).

Infant colics were further classified as severe if the episodes occurred more than twice a day. The diary served as an indicator for the need of a medical examination, and as a consistent way of recording and recalling symptoms. Parents were instructed to contact the investigator if necessary, and to avoid the use of prebiotics, probiotics, symbiotics and immune-stimulating products during the 3-month study period.

**Assessment of immunological parameters.** Faecal samples were collected and immediately frozen at −20 °C, and then stored at −80 °C at our institution. For secretory immunoglobulin A (sIgA) and beta-defensin 2 (HBD-2), 1 g of faecal sample was diluted 1:1 (w/v) with PBS buffer (130 mM NaCl and 10 mM sodium phosphate-buffered saline, pH 7.4). For alpha-defensins (HNP 1–3): 1 g of faecal sample was diluted 1:0.5 (w/v) with the same buffer. This was necessary considering the detection limit of the kit that was commercially available when the investigation was performed (i.e., 0.05 ng/g). All samples were then centrifuged at 13,000 rpm for 15 min in 1.5-ml tubes. The supernatant was collected for quantification by ELISA, without further dilution. HNP 1–3 was measured by ELISA using a specific human kit (Hycult Biotechnology, Uden, The Netherlands), HBD-2 by ELISA using a specific human kit (Phoenix Pharmaceuticals, Inc., Burlingame, CA, USA) (detection limit: 0.01 ng/g) and sIgA by indirect enzyme immunoassay for human samples (Salimetrics LLC, Carlsbad, CA, USA) (detection limit: 2.5 μg/g)[32]. For LL-37 measurement, the sample (1 g of faecal sample) was extracted with 60% acetonitrile in 1% aqueous trifluoroacetic acid (TFA) and then extracted overnight at 4 °C[53]. The extract was centrifuged, and the supernatant stored at −20 °C. LL-37 level was then measured, without dilution, by a commercially available ELISA kit specific for human samples (Hycult Biotechnology, Uden, The Netherlands) (detection limit: 0.1 ng/g).

**Microbiota analysis.** Microbiota analysis was performed on 9 newborns per group, and measurements were acquired at enrolment (T0) and visit 2 (T2). DNA was extracted from 200 mg of faecal samples with G'NOME DNA isolation kit (MP Biomedicals) following a published protocol[54]. Briefly, faecal samples were homogenised in the supplied cell suspension buffer. RNase mix and Cell lysis/denaturing solution were then added, and the samples incubated at 55 °C for 30 min. Protease mix was added, and the samples incubated at 55 °C for 2 h. To improve cellular lysis, 750 μL of 0.1-mm-diameter silica beads were added, and samples were put in a Beadbeater (MP Biomedicals) (ten cycles at maximum speed for 45 s). Polyvinylpolypyrrolidone (15 mg) was added, and the samples were vortexed and centrifuged at 20,000 g for 3 min. The supernatant was recovered. The remaining pellet was washed with 400 μL of TENP [50 mM Tris (pH 8), 20 mM EDTA (pH 8), 100 mM NaCl and 1% polyvinylpolypyrrolidone] and centrifuged at 20,000 g for 3 min. The washing step was repeated twice more, and the resulting supernatants pooled. The supernatant collected was centrifuged at 20,000 g for 3 min, and 750 μL were transferred in a clean 2-mL microcentrifuge tube. Nucleic acids were precipitated by addition of 1 ml of isopropanol, storage at −20 °C for 20 min and centrifugation at 20,000 g for 5 min. The pellet was dried with the speed vacuum, and then resuspended in 400 μL of distilled water plus 100 μL of salt-out mixture, and incubated at 4 °C for 10 min. Samples were centrifuged at 20,000 g for 10 min, and the supernatant containing the DNA was transferred to a clean 1.5-mL microcentrifuge tube. DNA was precipitated with 1.5 mL of 100% ethanol at room temperature for 5 min, followed by centrifugation at 20,000 g for 5 min. Residual ethanol was removed using speed vacuum. DNA was resuspended in 150 μL of water and stored at −20 °C. Partial 16S rRNA gene sequences were amplified from extracted DNA using primer pair Probio_Uni/Probio_Rev, which target the V3 region of the 16S rRNA gene sequence (see Supplementary Table 4)[55]. 16S rRNA gene amplification and amplicon checks were carried out, and 16S rRNA gene sequencing was performed using a MiSeq (Illumina) at the DNA sequencing facility of GenProbio srl[55]. Following sequencing, the.fastq files were processed using the QIIME2 software suite[56]. Quality control was performed with the DADA2 pipeline, and reads were truncated at the first instance of a quality score less than or equal to 20. A phylogenetic tree was built using fasttree and mafft alignment, and diversity measures (alpha- and β-diversity indices) were calculated. Community richness (alpha-diversity) was evaluated by Chao1 index and represented by box-and-whisker plot. Community dissimilarities (β-diversity) were quantitatively evaluated by weighted UniFrac distance[57] and represented by a PCoA plot. Q2-feature-classifier, trained on the SILVA132 99% OTUs[58], specifically on the V3 region, was used to perform taxonomic classification. Differential abundance of bacterial genera was tested by ANOVA and TukeyHSD post hoc analysis for multiple comparison ($P < 0.05$). Microbiota taxa showing different abundance trends (T0–T2) according to delivery–diet groups were identified with ANOVA. The analysis accounted for the individual baseline of each subject, and for fixed effects of diet and delivery using a linear mixed model (lmerTest R package). We selected microbiota showing a significant interaction between the delivery–diet groups and time ($P < 0.05$).

**Metabolite extraction and derivatisation.** Metabolome extraction from faecal samples, purification and derivatisation were carried out by means of the MetaboPrep kit (Theoreo srl, Montecorvino Pugliano [SA], Italy) according to the manufacturer's instruction.

**Gas chromatography mass spectrometry analysis.** Two-microlitre samples of the derivatised solution were injected into the GC–MS system (GC-2010 Plus gas chromatograph coupled to a 2010 Plus single-quadrupole mass spectrometer, Shimadzu Corp., Kyoto, Japan). Chromatographic separation was achieved with a 30-m 0.25-mm CP-Sil 8 CB-fused silica capillary GC column with 1.00-μm film thickness from Agilent (Agilent, J&W Scientific, Folsom, CA, USA), with helium as carrier gas. The initial oven temperature of 100 °C was maintained for 1 min, and then raised by 4 °C/min to 320 °C with a further 4 min of hold time. The gas flow was set to obtain a constant linear velocity of 39 cm/s, and the split flow was set at 1:5. The mass spectrometer was operated in electron impact (70 eV) in full-scan mode in the interval of 35–600 m/z with a scan velocity of 3333 amu/s and a solvent cut time of 4.5 min. The complete GC programme duration was 60 min. Untargeted metabolites were identified by comparing the mass spectrum of each peak with the NIST library collection (NIST, Gaithersburg, MD, USA). To identify metabolites' identity, the linear index difference maximum tolerance was set at 10, while the minimum matching for the NIST library search was set at 85%. All measurements were performed by Theoreo srl, Montecorvino Pugliano (SA), Italy.

**Direct injection Fourier-transform ion cyclotron resonance mass spectrometry analysis.** Analyses were performed in direct infusion employing a Hamilton syringe (250 μL) at a flow rate of 2 μL/min. Data were acquired on a SolariX XR 7 T (Bruker Daltonics, Bremen, Germany). The instrument was tuned with a standard solution of sodium trifluoroacetate (NaTFA). Mass spectra were recorded in broadband mode in the range 100–1500 m/z, with an ion accumulation of 20 ms, with 32 scans using 2 million data points (2 M). Nebulising (nitrogen) and drying (air) gases were set at 1.0 and 4.0 mL/min, respectively, with a drying temperature of 200 °C. The instrument was controlled by Bruker FTMS Control, MS spectra were elaborated with Compass Data Analysis version 4.2 (Bruker Daltonics, Bremen, Germany) and identification of compounds based on accurate MS measurements was performed by Compound Crawler ver. 3.0 (Bruker).

**Adverse events.** Adverse events were recorded throughout the study period. They were assessed based on inquiries to the parents and on daily records. An adverse event was defined as any event that was not consistent with the information provided in the consent form, or that could not reasonably be expected to accompany the natural history and progression of the subject's condition throughout the study. All adverse events were evaluated by the investigator for causal relationship to the study feeding and for severity. Adverse events were considered serious if they were fatal or life-threatening, required hospitalisation or surgical intervention, resulted in persistent or significant disability/incapacity or were considered to be medically relevant by the investigator. All other adverse events were categorised as non-serious.

**Statistical analysis.** The sample size was determined to identify a difference in the content of immunological peptides on faecal samples between the infants fed formula F and infants fed formula S. The breastfed infants constituted the reference group. The power of the study was calculated based on the result of previous clinical trials where fermented milk with *L. paracasei* CBA L74 was able to significantly stimulate the intestinal production of sIgA and innate immunity peptides in young children attending preschool centres[31,32]. With a study power of 80% and an alpha level of 0.05, we calculated to enrol 13 newborns per each group. Assuming a 30% drop-out rate, at least 16 newborns per group must be recruited.

Continuous variables are reported as mean and standard deviation (SD) or median and range. Categorical variables are reported as absolute numbers or percentages. Differences between groups in terms of growth, body composition and immunological parameters, were assessed by analysis of variance. Comparisons among groups were performed using the $\chi^2$ test for comparisons between discrete variables. Differences in continuous variables (growth, body composition and immunological parameters) between groups were assessed by analysis of variance, while $\chi^2$ test was used for comparisons between categorical variables.

For the analysis of the main outcome, two-way ANOVA with TukeyHSD post hoc test and Kruskal–Wallis test was used. A repeated measures Friedman nonparametric test was used to evaluate the modification of the immunological parameters throughout the study.

Statistical significance was set at $\alpha = 0.05$. Statistical analyses were conducted using SPSS (Statistical Package for the Social Sciences) version 12 software (SPSS Inc., Chicago, IL, USA). Linear mixed model (lmerTest R package) and ANOVA were used to test the interaction between group and time in order to identify significantly different trajectories of IgA levels according to groups and time. Emmeans R package was used to perform TukeyHSD post hoc test.

We included the statistical analysis to the microbiota in the section 'Microbiota analysis'.

Regarding the metabolome, analysis was performed on the same newborns of the microbiota analysis (9 newborns per group), and measurements were acquired

at enrolment (T0) and visit 2 (T2). The chromatographic data for Partial Least Squares Discriminant Analysis (PLS-DA) were tabulated with one sample per row and one variable (metabolite) per column. According to MSI level 1 standard[59], the VIP putative metabolite identity was confirmed by means of an independent analytical standard analysis. The normalisation procedures consisted of data transformation and scaling. Data transformation was made by generalised log transformation and data scaling by autoscaling (mean-centered and divided by standard deviation of each variable). Metabolites showing different concentrations across the diet and delivery groups were identified separately at T0 and T2 with ANOVA (FDR < 0.05) and Fisher's least significant difference (LSD) test for pairwise comparisons ($P < 0.05$). The microbe–metabolite correlation was evaluated by means of a heatmap that was created by calculating Spearman correlation coefficients for each pairwise combination of microbial genus abundances and metabolite intensities using the 'corr.test' function in the 'psych' (version 1.8.12) R package [https://cran.r-project.org/package=psych]. Metabolites showing different concentration trends (T0–T2) according to delivery–diet groups were identified with ANOVA. The analysis accounted for the individual baseline of each subject, and for fixed effects of diet and delivery using a linear mixed model (lmerTest R package). We selected metabolites showing a significant interaction between the delivery–diet groups and time ($P < 0.05$).

**Reporting summary**. Further information on research design is available in the Nature Research Reporting Summary linked to this article.

## Data availability
Data supporting the findings of this study are available within the paper and its Supplementary Information files. Data underlying Figs. 2, 3a, c, Supplementary Figs. 2–5 are provided as Source Data files. All other data are available from the corresponding author upon reasonable requests. The data sets generated and analysed during this study regarding the 16S rRNA microbiota analysis are available in the Sequence Read Archive repository (SRA accession: PRJNA608934); those regarding the metabolome analysis are identified in MetaboLights as MTBLS1532 [https://www.ebi.ac.uk/metabolights/MTBLS1532].

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

## Acknowledgements

We thank the study participants. M.R. is supported by grants from the Italian Association for Cancer Research (AIRC IG 17628) and the European Research Council (No. 615735—homeogut ERC). This work was supported by Kraft-Heinz.

## Author contributions

Conceptualisation: M.R., F.M. and P.R.; methodology: N.L., C.P., J.T., D.B., C.M., L.P. and A.B.; data interpretation and analysis: C.P., D.B., J.T. and R.N.; writing, reviewing and paper editing: M.R., C.P., D.B., ML.G., P.R. and R.B.C.; funding: M.R. and F.M.

## Competing interests

This work was sponsored by Heinz Italia S.p.A. As an employee of Heinz Italia S.p.A., A. B. has contributed technologically in adapting the fermentation process to the generation of the fermented formula for infant use, and the sponsor has provided the two formulas (standard and fermented) for the study. Moreover, the sponsor has contributed funding for biological assays, including ELISA, kits and –OMICS analyses. A.B. did not play any role in study design, analysis or writing of the paper (except for the methodological section on the formulas). The remaining authors declare no competing interests.
