## [Peer Review File · Nature Communications]

Reviewers' comments:

Reviewer #1 (Remarks to the Author):

In this pilot study, authors performed a randomized controlled clinical trial to compare immune responses (antimicrobial peptides, IgAs), microbiota and metabolome, in infants fed a *Lactobacillus paracasei* CBA L74 fermented formula), formula alone or breast milk. The formula babies did not show the normal increase in secretory IgAs observed in breast milk and in less degree in the lactobacillus supplemented babies, and showed higher microbiota diversity than normal. Bifidobacteria was more associated to vaginal birth than to milk type, and metabolites discriminated well the baby groups.

This is an elegant small pilot study showing that *Lactobacillus paracasei* CBA L74 supplementation of formula milk partially normalizes immune and microbiota early development.

Comments and suggestions:

- 1- What was the age range of enrollment?
- 2- Please clarify, were the maternal milk group breastfed, or bottle-fed expressed maternal milk, or both?
- 3- Gestational age varied widely from 37-41 weeks. This would affect birth weight and other factors. Was there any analyses performed to control for this? Can you comment in the paper?
- 4- Since the N is small in the baby groups, the should present the statistical power and discuss the sample desirable in a bigger clinical trial.
- 5- It would have been ideal if they can follow these babies over the first year of life, at least. If they plan to, they should mention it in discussion.
- 6- Figures should be self contained; in Fig 4, panels of PCA, the legend could be more explicit, and in the caption of the figure, all groups should be spelled out, including formula D and formula M

Reviewer #2 (Remarks to the Author):

This manuscript outlines a detailed study to compare breast feeding, standard formula and *Lactobacillus paracasei* fermented infant formula on microbiome development in full-term human infants. Focus is given to examine changes in the microbial population and their effects on immune parameters of the human host. The authors provide metadata ranging from infant characteristics, birth-type, microbiota analysis with 16S, and metabolite detection with GC-MS. They focus on detailed results that compare and contrast the feeding types and the resulting effects.

In general, this study is scientifically sound and logically designed. The authors have used an appropriate population dataset, and have tried to control (or at least understand) the possible confounding variables. Even with all this work, the results are a bit underwhelming and fairly consistent with what is already known. That being said, there are some informative results about fermented formula feedings.

Some specific comments are listed below:

1. The title could be made more descriptive, to provide a “so-what-is-learned” nugget to better engage the reader.
2. The focus on only 16S taxa profiling is far too limited. While this has been the standard and does provide interesting information, current state-of-the-art expectations include metagenome sequencing/assembly for most microbiome systems. 16S does not reveal the full range of possible novel taxa that would be revealed by deeper de-novo sequencing, and therefore may be too limited to capture the range and complexity of human microbiome systems.
3. Microbiota analysis only by “population counting” is too limited, and does not reveal anything about the metabolic activities of the microbiota. Which ones are present, but inactive? Who are the major drivers of metabolic function? What are the major microbiota activities and how do these differ with the different conditions? To only use 16S is analogous to glancing into a conference room and reporting that “10 people of age 50-60 are present, but without any knowledge of what they are doing or talking about.” The authors should at least have had transcriptomics and/or metaproteomic information.
4. The authors do provide some limited metabolite information, but there is too big of a gap between the microbiota “head-count” and the metabolite response. This is not to say that there

results are inaccurate, but they miss too much of the critical “in-between-information” about the metabolic activities of the microbiota and how they differ between conditions.

5. Based on the points raised above, I find this to be a “broadly descriptive” manuscript without adequate drill-down into a deeper understanding. Thus, this paper is informative and may be appropriate for another journal, but misses the expectation marks for this journal.

Reviewer #3 (Remarks to the Author):

This RCT compared biomarkers of immune function, gut microbiota composition, and gut metabolome between infants receiving standard formula or a new fermented formula, with breastfed infants as a reference group. This is a very interesting and clinically important study design; however, the execution suffers from several major problems. The biological and statistical analyses are overly simplistic, the presentation of results is ineffective and confusing, the paper is not well written, and the interpretation and conclusions are often unsubstantiated by the data. I commend the authors for undertaking this important study, but a major overhaul is needed to clearly determine and interpret the effects of this intervention. Some specific concerns are listed below.

1. The lack of information on the diet of infants in the first week before enrolment makes the baseline results confounded and hard to interpret. Many differences were observed at baseline for microbiota, metabolome, and other measured components. This is a major limitation and needs to be taken into account when examining other time points. It might help to analyze changes from baseline for each infant.

2. Breastfeeding is the biological norm, therefore you should not say that breastmilk “reduces” or “increases” various things. This should be rephrased throughout.

3. The trial registration indicates that 96 infants were enrolled; only 74 are mentioned in this study and even fewer were included in the microbiome analysis. Why?

4. Introduction: Generally, many of the references are out of date and the current literature on this topic is not well cited. Authors state that the mammary gland is colonized by microbiota; this is not confirmed and remains an open topic of investigation. Oligosaccharides are discussed but there is no clear distinction between human milk oligosaccharides, which are different from the synthetic oligosaccharides added to most formulas. "Postbiotics" should be defined. The rationale for stratification based on mode of delivery is not clearly explained.

5. General methods: "Randomized to receive breast milk" – this would be unethical; please clarify or rephrase. Please briefly explain how the fermented formula was made, and why the particular strain was used. The naming of formulas (D for standard, M for fermented) is not intuitive. A full diagram showing the enrolment and follow up of participants should be included.

6. Microbiome methods: 16S amplicon sequencing is not a metagenomic analysis. Shannon is not a measure of richness, it is a measure of diversity taking into account evenness. The description of rarefaction is unusual. Unifrac is a measure of distance not similarity. Pearson is not a distance metric. Many microbiome analyses described are not reported in the results.

7. Results: Overall, the presentation of results is poor. The tables are formatted in various inconsistent ways; panels within the same figure use different scales, formats, ordering of study groups, and colour schemes; colour is not used effectively. Some of the results described in the text are not supported by figures and conclusions throughout are very superficial. The middle timepoint is often not shown – why?

8. Discussion: many claims are unsubstantiated by the results. Discussion on Bifidobacterium is especially misleading as the majority of samples did not contain Bifidobacterium. There is no discussion of study limitations.

We would like to thank the reviewers for their very helpful suggestions that we think have helped improve our manuscript. Below a point-by-point response to reviewers' comments.

Reviewers' comments:

Reviewer #1 (Remarks to the Author):

In this pilot study, authors performed a randomized controlled clinical trial to compare immune responses (antimicrobial peptides, IgAs), microbiota and metabolome, in infants fed a Lactobacillus paracasei CBA L74 fermented formula), formula alone or breast milk. The formula babies did not show the normal increase in secretory IgAs observed in breast milk and in less degree in the lactobacillus supplemented babies, and showed higher microbiota diversity than normal. Bifidobacteria was more associated to vaginal birth than to milk type, and metabolites discriminated well the baby groups.

This is an elegant small pilot study showing that Lactobacillus paracasei CBA L74 supplementation of formula milk partially normalizes immune and microbiota early development.

We thank the reviewer for the appreciation on our study.

Comments and suggestions:

1- What was the age range of enrollment?

The age range at enrolment was 0-7 days of life. The mean age at enrolment was 3.3 ± 1.6 days of life. This is now clarified in the main text.

2- Please clarify, were the maternal milk group breastfed, or bottle-fed expressed maternal milk, or both?

The reference group was constituted by breastfed infants. No infants were bottle-fed. This is now clarified in the methods section.

3- Gestational age varied widely from 37-41 weeks. This would affect birth weight and other factors. Was there any analyses performed to control for this? Can you comment in the paper?

We included in the study only full-term infants, all with birth weight adequate for their gestational age. As the mean weight and gestational age at birth were similar among groups, we did not perform any correction. We have now added a sentence in the main text of the paper.

4- Since the N is small in the baby groups, the should present the statistical power and discuss the sample desirable in a bigger clinical trial.

The sample size of this study was calculated according with the values of fecal sIgA based on the results of a previous study on the effect of a treatment with milk fermented with L. paracasei CBA L74 in children attending preschool center. Please, see reference n. 26, 27.

The power of this study was 80% and an alpha of 0.05, therefore we calculated to enroll 13 newborns per each group, excluding the drop out. The study has achieved its primary objective as we observed a statistically significant difference between standard and fermented formula in sIgA in the feces. We clarified this point.

In order to design a bigger clinical trial, a different variable than the fecal sIgA has to be taken into consideration.

5- It would have been ideal if they can follow these babies over the first year of life, at least. If they plan to, they should mention it in discussion.

Thank you for the interesting suggestion. It would be interesting to conduct a follow up study on the clinical outcomes of these infants. Unfortunately, it was not foreseen, but we may think of asking for an amendment.

6- Figures should be self contained; in Fig 4, panels of PCA, the legend could be more explicit, and in the caption of the figure, all groups should be spelled out, including formula D and formula M. The reviewer is right, we have now used the same symbols and colors throughout the study to make it simpler to make comparisons. We have also changed the legend and shortened the different diets to S: standard and F: fermented. We abandoned the previous nomenclature that was the one used during the randomization.

Reviewer #2 (Remarks to the Author):

This manuscript outlines a detailed study to compare breast feeding, standard formula and *Lactobacillus paracasei* fermented infant formula on microbiome development in full-term human infants. Focus is given to examine changes in the microbial population and their effects on immune parameters of the human host. The authors provide metadata ranging from infant characteristics, birth-type, microbiota analysis with 16S, and metabolite detection with GC-MS. They focus on detailed results that compare and contrast the feeding types and the resulting effects.

In general, this study is scientifically sound and logically designed. The authors have used an appropriate population dataset, and have tried to control (or at least understand) the possible confounding variables. Even with all this work, the results are a bit underwhelming and fairly consistent with what is already known. That being said, there are some informative results about fermented formula feedings.

We thank the reviewer for the positive comments. We have now added new comparisons to take into account both the microbiota and the metabolome so to exploit as much as possible the data.

Some specific comments are listed below:

1. The title could be made more descriptive, to provide a “so-what-is-learned” nugget to better engage the reader.

Yes, we have changed the title to engage more the reader.

2. The focus on only 16S taxa profiling is far too limited. While this has been the standard and does provide interesting information, current state-of-the-art expectations include metagenome sequencing/assembly for most microbiome systems. 16S does not reveal the full range of possible novel taxa that would be revealed by deeper de-novo sequencing, and therefore may be too limited to capture the range and complexity of human microbiome systems.

We agree with the reviewer, but the major aim of this project was to evaluate whether the fermented formula was a better substitute for breast milk than the standard formula to offer a better chance to children that could not be breast fed. This is why we chose to perform 16S rRNA analysis. In addition, the amount of feces obtained in the newborns were quite variable especially at enrolment and for some samples we had really just enough for 16S and metabolomics, we could not perform anything else for many newborns. So this is not feasible now as we would have to further reduce the number of samples as we would miss several samples at enrolment.

3. Microbiota analysis only by “population counting” is too limited, and does not reveal anything about the metabolic activities of the microbiota. Which ones are present, but inactive? Who are the major drivers of metabolic function? What are the major microbiota activities and how do these differ with the different conditions? To only use 16S is analogous to glancing into a conference room and reporting that “10 people of age 50-60 are present, but without any knowledge of what they are doing or talking about.” The authors should at least have had transcriptomics and/or metaproteomic information.

We have now performed a combined analysis of 16S rRNA and metabolome to correlate some metabolites with the microbiota enriched in the different diets. Interestingly, we observed a cluster of metabolites that positively or negatively correlated with bacterial genera, in some cases, more abundant in the different groups (particularly breast-feeding and standard formula). The fermented formula does not seem to be characterized by a particular set of microbes and metabolites that separates from the other two groups, confirming that this diet generates a situation that is really in between the two dietary regimens.

4. The authors do provide some limited metabolite information, but there is too big of a gap between the microbiota “head-count” and the metabolite response. This is not to say that there results are inaccurate, but they miss too much of the critical “in-between-information” about the

metabolic activities of the microbiota and how they differ between conditions.

As mentioned above, to fill the gap we performed a complex combined metabolome-microbiota analysis. We identified sets of metabolites that were enriched in the different diets and that correlated with sets of taxa from the microbiome at genus level.

5. Based on the points raised above, I find this to be a “broadly descriptive” manuscript without adequate drill-down into a deeper understanding. Thus, this paper is informative and may be appropriate for another journal, but misses the expectation marks for this journal.

We hope that with the new analyses we have convinced the reviewer that there are important take home messages for this manuscript that make it of a broad interest as that of Nature communications. In particular, we have demonstrated that the metabolome is a better tool to analyse small sample groups. We have observed that sets of metabolites clusterize and correlate with the different diet groups. We have correlated these sets of metabolites that were positively or negatively enriched in the different diets with particular groups of genera enriched in the same diets. We think that this manuscripts sets the basis for future studies.

Reviewer #3 (Remarks to the Author):

This RCT compared biomarkers of immune function, gut microbiota composition, and gut metabolome between infants receiving standard formula or a new fermented formula, with breastfed infants as a reference group. This is a very interesting and clinically important study design; however, the execution suffers from several major problems. The biological and statistical analyses are overly simplistic, the presentation of results is ineffective and confusing, the paper is not well written, and the interpretation and conclusions are often unsubstantiated by the data. SI commend the authors for undertaking this important study, but a major overhaul is needed to clearly determine and interpret the effects of this intervention. Some specific concerns are listed below.

We thank the reviewer for this constructive remark, we think that we have addressed his/her concerns.

1. The lack of information on the diet of infants in the first week before enrolment makes the baseline results confounded and hard to interpret. Many differences were observed at baseline for microbiota, metabolome, and other measured components. This is a major limitation and needs to be taken into account when examining other time points. It might help to analyze changes from baseline for each infant.

We enrolled infants before 7 days of life. Specifically, the mean age at enrolment was 3.3 ± 1.6 days of life. Before the enrolment the infants of the reference group were exclusively breastfed, whereas infants randomized on standard (S) formula or fermented (F) formula received all the same standard (S) formula since birth to the enrolment. Indeed, we selected infants to be randomized from mothers that had a priori decided not to breast feeding. Hence, the differences between breast feeding and the other two regimens are observed already at enrolment because babies have already breastfed, changes in formula fed are visible only at T2 and are the consequence of the diets. We have clarified this in the manuscript. Unfortunately we could not do differently as it takes a few days to enrol the children.

2. Breastfeeding is the biological norm, therefore you should not say that breastmilk “reduces” or “increases” various things. This should be rephrased throughout.

We have corrected throughout.

3. The trial registration indicates that 96 infants were enrolled; only 74 are mentioned in this study and even fewer were included in the microbiome analysis. Why?

The trial registration indicates 96 infants because we included the drop-out rate. Indeed, it was described in the statistical analysis section *“Based on the results of a previous study on the effect of a treatment with milk fermented with L. paracasei CBA L74 on fecal sIgA in children attending preschool centers, a study power of 80% and an alpha of 0.05, we calculated to enroll 13 newborns per each group. Assuming a 30% drop-out rate, at least 16 newborns per group must be recruited”*. Therefore, 74 was the number of infants that completed the follow up. We have clarified this point.

4. Introduction: Generally, many of the references are out of date and the current literature on this topic is not well cited. Authors state that the mammary gland is colonized by microbiota; this is not confirmed and remains an open topic of investigation. Oligosaccharides are discussed but there is no clear distinction between human milk oligosaccharides, which are different from the synthetic oligosaccharides added to most formulas. “Postbiotics” should be defined. The rationale for stratification based on mode of delivery is not clearly explained.

Regarding colonization of the mammary gland, this has been described in non-human primates and other lower organisms (mice), hence we gave it for granted that it was the same in humans, but the reviewer is right, hence we have rephrased the sentence to make it clear that this has not been demonstrated. We have better described what are postbiotics and have updated the literature. We have also explained the rationale for stratifying against the mode of delivery.

5. General methods: “Randomized to receive breast milk” – this would be unethical; please clarify or rephrase. Please briefly explain how the fermented formula was made, and why the particular strain was used. The naming of formulas (D for standard, M for fermented) is not intuitive. A full diagram showing the enrolment and follow up of participants should be included.

We are sorry for this statement. As the reviewer points out only the infants that would be receiving the formula have been randomized. This is now clarified in the text. A diagram of the enrolment and follow up is now shown in supplementary figure 1. We have changed the naming for the different diets to S: standard and F: fermented. We abandoned the previous nomenclature that was the one used during the randomization.

6. Microbiome methods: 16S amplicon sequencing is not a metagenomic analysis. Shannon is not a measure of richness, it is a measure of diversity taking into account evenness. The description of rarefaction is unusual. Unifrac is a measure of distance not similarity. Pearson is not a distance metric. Many microbiome analyses described are not reported in the results.

The methods section on the microbiota has been corrected

7. Results: Overall, the presentation of results is poor. The tables are formatted in various inconsistent ways; panels within the same figure use different scales, formats, ordering of study groups, and colour schemes; colour is not used effectively. Some of the results described in the text are not supported by figures and conclusions throughout are very superficial. The middle timepoint is often not shown – why?

The tables and figures have been reformatted so to unify all the groups with the same colors and symbols. In some cases we could not unify the scales because we would miss differences among groups. The middle time point was only used for clinical assessment. This is why it is not reported for biological evaluations (IgAs, defensins, microbiome and metabolome).

8. Discussion: many claims are unsubstantiated by the results. Discussion on Bifidobacterium is especially misleading as the majority of samples did not contain Bifidobacterium. There is no discussion of study limitations.

The part on Bifidobacterium is now substantiated by a better representation (taxa barplot) and numbers of samples positive or negative (the negative samples are primarily in the feces of infants born by caesarean section). Study limitations are thoroughly discussed now, thank you.

Reviewers' comments:

Reviewer #1 (Remarks to the Author):

The authors have done a good job responding to the reviewers suggestions.

My only concern is the statement in the introduction that the mammary gland has a microbiome. As with the placenta, non-epithelial internal organs are immunoprotected, and are expected, in healthy people with healthy immune systems, to be sterile.

We have learnt from the story on the "placental microbiome" that it is misleading to attribute the 16S DNA sequences to an organ that has very low 16S DNA load, and that this is due to contamination. Even if there was a few sequences, the food can also contain temporarily a low bacterial load, but by no means this indicates the existence of a blood microbiome. The cited work of Shively et al does not clearly state the nature of the breast specimens. It is expected that, like the urethra, or the ear, there is a gradient towards sterility from the outer to the inner canal, but the dogma of internal sterility in health, with a sterilizing immune system, remains valid.

Thus, I agree with reviewer three that a mammary gland microbiome cannot be taken for granted.

Reviewer #2 (Remarks to the Author):

The authors have done a reasonable job in responding to the three sets of review comments. Unfortunately, most of the serious concerns from reviewers #2 and 3 are not addressed in adequate detail. While the authors have added a bit more metabolomics data, this entire study is still too limited in experimental approach, and the discussion is too broad and vague.

While this is an important study, the quality and depth are still not at an adequate level appropriate for this journal.

Reviewer #3 (Remarks to the Author):

Overall, it was frustrating to review this revised manuscript. The authors did not clearly indicate the revisions in their responses (i.e. did not quote the new additions or refer to specific page/line

numbers or figures). Some improvements have been made (e.g. the figure formatting is much better), but overall I still have many concerns. While this is an interesting study of potential interest for another journal, I do not think it meets the bar for Nature Communications.

A few comments, which are not exhaustive:

- I suggested analyzing changes from baseline for each infant. This was not done and no justification was provided for why not.
- Authors stated that they corrected their language throughout (in response to my comment that 'reduce' and 'increase' are not appropriate when referring to breastmilk, which is the biological norm). This is still misused throughout.
- In the introduction, the section on prebiotics/oligosaccharides is still unclear. Prebiotics can include GOS/FOS or synthetic HMOs. This is not clearly described.
- Authors state they have updated the literature review but they have not highlighted the new references.
- I commented that "Some of the results described in the text are not supported by figures and conclusions throughout are very superficial." – The author response does not address this.
- The new taxa barplot is completely unhelpful for assessing Bifidobacteria because a gradient colour scheme is used and I cannot tell Bifidobacteria apart from Bacteroides.

We would like to thank the reviewers for their helpful suggestions that we think have further improved our manuscript. Below a point-by-point response to reviewers' comments.

We have now:

- 1- Performed two new analyses which take into account the difference of each infant at baseline as per reviewer #3 suggestion (reported in New Extended data Fig. 4 for the microbiota and new Extended data Fig. 5 for the metabolites)
- 2- We have rewritten the discussion to include additional sessions to discuss the different metabolites and microbiota strains identified as characterizing the different dietary regimens and their possible implications
- 3- Clearly described which changes have been made and where (in the detailed point-by-point below).
- 4- Highlighted in yellow/underlined the new corrections keeping those only underlined as per the previous revision
- 5- Rephrased and corrected the language 'increased or decreased' relative to the reference group
- 6- Modified the representation of the Bifidobacteria to make it clearer that the difference at T0 is primarily linked to the mode of delivery and secondary to breastfeeding (new panel Fig. 3C)

Reviewers' comments:

Reviewer #1 (Remarks to the Author):

The authors have done a good job responding to the reviewers suggestions. My only concern is the statement in the introduction that the mammary gland has a microbiome. As with the placenta, non-epithelial internal organs are immunoprotected, and are expected, in healthy people with healthy immune systems, to be sterile.

We have learnt from the story on the "placental microbiome" that it is misleading to attribute the 16S DNA sequences to an organ that has very low 16S DNA load, and that this is due to contamination. Even if there was a few sequences, the food can also contain temporarily a low bacterial load, but by no means this indicates the existence of a blood microbiome. The cited work of Shively et al does not clearly state the nature of the breast specimens. It is expected that, like the urethra, or the ear, there is a gradient towards sterility from the outer to the inner canal, but the dogma of internal sterility in health, with a sterilizing immune system, remains valid.

Thus, I agree with reviewer three that a mammary gland microbiome cannot be taken for granted.

We thank the reviewer for the positive comment. We completely agree that recent findings indicate that the placenta is sterile and that what previously described is likely to be an artifact, however, the identification of a microbiota in the mammary gland has not been either demonstrated or confuted. It is unlikely that the microbial DNA found in the mammary gland is the consequence of a contamination as samples from breast cancer patients have shown a different DNA composition when analyzing the healthy or the neoplastic tissue. (Urbaniak, C.; Gloor, G.B.; Brackstone, M.; Scott, L.; Tangney,

M.; Reid, G. The Microbiota of Breast Tissue and Its Association with Breast Cancer. *Appl. Environ. Microbiol.* 2016, 82, 5039–5048. [CrossRef] [PubMed]; Banerjee, S.; Wei, Z.; Tan, F.; Peck, K.N.; Shih, N.; Feldman, M.; Rebbeck, T.R.; Alwine, J.C.; Robertson, E.S.; Distinct microbiological signatures associated with triple negative breast cancer. *Sci. Rep.* 2015, 5, 15162. [CrossRef]; Chan, A.A.; Bashir, M.; Rivas, M.N.; Duvall, K.; Sieling, P.A.; Pieber, T.R.; Vaishampayan, P.A.; Love, S.M.; Lee, D.J. Characterization of the microbiome of nipple aspirate fluid of breast cancer survivors. *Sci. Rep.* 2016, 6, 28061. [CrossRef]; Hieken, T.J.; Chen, J.; Hoskin, T.L.; Walther-Antonio, M.; Johnson, S.; Ramaker, S.; Xiao, J.; Radisky, D.C.; Knutson, K.L.; Kalar, K.R.; et al. The Microbiome of Aseptically Collected Human Breast Tissue in Benign and Malignant Disease. *Sci. Rep.* 2016, 6, 30751. [CrossRef]; Urbaniak, C.; Cummins, J.; Brackstone, M.; Macklaim, J.M.; Gloor, G.B.; Baban, C.K.; Scott, L.; O'Hanlon, D.M.; Burton, J.P.; Francis, K.P.; et al. Microbiota of human breast tissue. *Appl. Environ. Microbiol.* 2014, 80, 3007–3014. [CrossRef] [PubMed]. **If it were a contamination it would probably lead to the same composition. However, as the reviewer points out there is no clear demonstration that this bacterial DNA is actually associated to live bacteria. Thus, we should be cautious. Hence, we have modified the following paragraph in page 3 lines 68-75:**

'The mammary gland of non-human primates has been shown to harbor microbial DNA whose composition is regulated by the diet⁹. Similarly, in the mouse mammary gland, microbial DNA has been observed together with an Immunoglobulin (Ig)A-enriched immune response¹⁰. Although the presence of microbial DNA does not necessarily indicate that the mammary gland is colonized by an indigenous microbiota, the finding that breast milk and infant gut harbor microbial strains characterized by antibiotic resistance genes (ARGs) identical to those found in their own mothers' gut microbiome suggests that during pregnancy or lactation, gut microbes or their constituents could translocate to the mammary gland^{11,12}. Consistently, in non-human primates the mammary gland has been shown to harbor bile acid analogs and microbial bioactive compounds which might be released in the milk⁹.'

Reviewer #2 (Remarks to the Author):

The authors have done a reasonable job in responding to the three sets of review comments. Unfortunately, most of the serious concerns from reviewers #2 and 3 are not addressed in adequate detail. While the authors have added a bit more metabolomics data, this entire study is still too limited in experimental approach, and the discussion is too broad and vague.

While this is an important study, the quality and depth are still not at an adequate level appropriate for this journal.

We are glad that the reviewer thinks that this is an important study. We hope that with the new revision, the reviewer is convinced that we have improved it and it is now acceptable for Nature Communication. We have tried to discuss more the different microbial and metabolite enrichments in the dietary regimens. Given the little knowledge on the activities of most of the identified metabolites, particularly in infants, the reviewer will agree that it is very difficult to speculate on their function and possible implications. This is just the first publication in which such a study has been undertaken and we hope the reviewer will appreciate the effort we have done in the new revision to go deeper in the discussion. Further studies will be needed to assess functionally (presumably in animal models) the implications of these findings. We have analyzed the differentially represented taxa as well as the metabolites, to try to give a meaning to the findings.

These sentences have been introduced in the discussion on page 17 lines 442-449:

'For example, we found that fermented formula fed infants had levels of myristic acid similar to those of breastfed infants, but reduced compared to standard formula fed infants. Myristic acid has been associated with the appetitive response in newborns³⁸. However, we did not observe any change in

body growth in the three different groups suggesting that its presence does not seem to influence food intake, at least in this time frame. Interestingly, intraperitoneal injection of myristic acid in a concentration similar to that found in the amniotic fluid to rats has been shown to induce an anxiolytic-like behaviour³⁹. Reduction of myristic acid in the fermented formula may give a health benefit to the newborn. ‘

And page 17 lines 450-462:

‘In particular, we observed a series of metabolites enriched in standard formula fed infants that were correlating with *Clostridium innocuum* group, *Erysipelotrichaceae*, *Eggerthella*, *Lachnospiraceae* and *Rumonicoccus 2* which were enriched in formula fed infants. Unfortunately, little is known on most of these bacteria and it is hard to infer possible advantages or disadvantages of having an enrichment of these bacteria. Ruminococci for instance have been shown to produce short chain fatty acids such as propionate and butyrate⁴⁰ that have several beneficial effects both on epithelial cell proliferation and barrier functions⁴¹, however there are no specific reports on *Rumonicoccus 2*. Similarly, regarding *Clostridium innocuum* there is only one report showing that it is associated with colobronchial fistula formation in Crohn’s disease together with other potential pathogens⁴². *Erysipelotrichaceae* also have been associated with inflammatory states of the gut⁴³, but again individual species might have different activities. More indepth analysis should be carried out to deepen our knowledge on the enrichment of these genera in standard formula fed infants.’

And pages 17-18 lines 465-481:

‘Keitoisocaproic acid is a metabolic intermediate of L-leucine while Leucine and Valine are branched aminoacids and are the bases for the production of other essential amino acids such as glutamate, glutamine, aspartate⁴⁴. It is not clear why in the formula fed group there should be increase of these aminoacids as they are highly represented in milk proteins⁴⁴. One possibility is that *Bacteroides*, *Parabacteroides* or *Veillonella* might use up these aminoacids or transform them thus reducing their concentration and hence since these taxa are not enriched in formula fed infants these aminoacids accumulate. Notably, *Bacteroides* make up a substantial portion of the human gut microbiota and are considered necessary to refine the gut environment and transformig it into one more hospitable for themselves and the other microorganisms⁴⁵. They reduce for instance oxygen levels intracellularly reducing at the same time both inflammatory reactive oxygen species and generating a strictly anaerobic environment that favors the growth of anaerobic microbes⁴⁶. In addition, *Bacteroides* are important for the digestion of complex molecules, including plant- or host-derived polysaccharides⁴⁵. As mentioned above, the human breast milk is rich in complex carbohydrates that are indigestible by our enzymes and this may explain the enrichment of *Bacteroides* in the reference group^{21,22}. *Veillonella*, on the other hand, uses lactate as its only source of carbon⁴⁷ and this may explain why lactose and lactulose are not abundant in the reference group.

Further, we have now added a new analysis of microbial strains or metabolites which are statistically significant among groups in each individual baby at time 0 and 3 months (T2). This analysis accounted for the individual baseline of each subject using a linear mixed model. Individual trends of microbiota at T0 and 3 months (T0-T2) are now reported in Fig. 3C regarding *Bifidobacterium*, in new supplementary Figure 4 for the other strains and in Extended data Figure 5 for the metabolites.

In new Fig. 3C it is clear that at enrollment in vaginally delivered babies (and even more so in breastfed infants) the level of bifidobacteria is higher than in caesarian delivered babies. This difference is lost at T2 (3 months) independent of the diet. This is reported in the text on page 9 lines 229-237

In Extended data Fig. 4 instead, we show all the reported strains that have demonstrated to be statistically significantly different when taking into account the individual differences at T0 among the babies. The analysis accounted for the individual baseline of each subject and for fixed effects of diet and delivery using a linear mixed model (lmerTest R package). As shown, there are some strains such as Eggerthella and Lachnospiraceae that were similarly regulated in both formulas fed caesarian delivered infants and differed from breast fed infants. Ruminococcus 2, Erysipelotrichaceae, gut metagenome, Clade I and partly also Ruminococcaceae UCG 014, were enriched in vaginally delivered standard formula fed infants at T2. This supports the difference in alpha diversity observed in vaginally delivered babies fed a standard diet. This is reported in the text on page 10 lines 257-268.

Finally, in Extended data Fig. 5, 20 statistically significant different metabolites in at least one group (by ANOVA $p < 0.05$) are reported for each individual infant over the course of the time (T0-T2). Interestingly, we observed that most of these metabolites that characterized the different groups were also clusterizing with particular sets of microorganisms in Fig. 5. This is reported in the text on pages 13-14 lines 311-339.

Reviewer #3 (Remarks to the Author):

Overall, it was frustrating to review this revised manuscript. The authors did not clearly indicate the revisions in their responses (i.e. did not quote the new additions or refer to specific page/line numbers or figures). Some improvements have been made (e.g. the figure formatting is much better), but overall I still have many concerns. While this is an interesting study of potential interest for another journal, I do not think it meets the bar for Nature Communications.

We are sorry that it was not easy to find in the revised manuscript the revisions according to the reviewers concerns. As we extensively revised the manuscript and highlighted the changes with underlined text, we did not report the exact positioning in the manuscript. Now we have highlighted the precise line and page where the revision has been carried out. In yellow the new revision while underlined the first revision.

A few comments, which are not exhaustive:

- I suggested analyzing changes from baseline for each infant. This was not done and no justification was provided for why not.

I think there was a misunderstanding in interpreting the request of the reviewer.

We thought that the reviewer was asking for clarification of information on the diet of infants in the first week before enrolment and that after knowing that the two analysed groups had received the same standard formula before randomization and that none of them had received breast milk, this was sufficient. Following the reviewer suggestion, we have now re-analysed the data taking into account also differences at baseline. The new analysis accounted for the individual baseline of each subject and for fixed effects of diet and delivery using a linear mixed model (lmerTest R package). Regarding the formula fed infants, the differences were even higher. We indeed found even more genera that were statistically significantly different as compared to the reference group. By contrast

we have lost some differences linked to the reference group longitudinally between T0 e T2. This could be due to quick changes in the microbiota that occurred already in the first days of breast feeding and that we cannot pinpoint as we cannot have time points before breast feeding.

This is now reported in Fig. 3C regarding Bifidobacterium and in new supplementary figure 4 for the other strains. In new Fig. 3C it is clear that in vaginally delivered babies (even more so for the reference group) the level of bifidobacteria is higher than in caesarian delivered babies. This difference is lost at T2 (3 months) independent of the diet.

This is reported in the text on page 9 lines 229-237 and reads as follows:

'The presence of Bifidobacteria was null in most of the samples except in vaginally delivered infants: breastfed (1 null out of 9), formula F fed infants (3 null out of 9) and formula S fed infants (4 null out of 9), indicating that vaginal delivery and early breastfeeding favours the enrichment of Bifidobacteria (Figure 3C). However, these differences were no longer significant at visit 2 where most of the infants displayed higher levels of Bifidobacteria even when analyzed per each individual subject (Figure 3C). This indicates that early colonization by Bifidobacteria is mostly related to the type of delivery; over time this group is similarly enriched independently of the mode of delivery or nutrition, even though breastfeeding favors the enrichment of Bifidobacteria early in life in vaginally delivered babies.'

In Extended data Fig. 4 instead, we show all the reported strains that have demonstrated to be statistically significantly different when taking into account the individual differences at T0 among the babies. The analysis accounted for the individual baseline of each subject and for fixed effects of diet and delivery using a linear mixed model (lmerTest R package). As shown, there are some strains such as Eggerthella and Lachnospiraceae that were similarly regulated in both formulas fed caesarian delivered infants and differed from breast fed infants. Ruminococcus 2, Erysipelotrichaceae, gut metagenome, Clade I and partly also Ruminococcaceae UCG 014, were enriched in vaginally delivered standard formula fed infants at T2. This supports the difference in alpha diversity observed in vaginally delivered babies fed a standard diet.

This is reported in the text on page 10 lines 257-268 and reads as follows:

'To exclude any possible confounding element derived from differences at enrolment linked to the initial feeding with standard diet versus breast milk we performed a statistical analysis which accounted for the individual baseline of each subject using a linear mixed model (lmerTest R package). Individual trends of the abundances of taxa at T0 and 3 months (T0-T2) that have demonstrated to be statistically significantly different by ANOVA in at least one group are shown in Extended data Fig. 4. This analysis allowed us to pin point more taxa that were differentially represented in the different groups than when the microbiome was analyzed in the whole population groups. Some taxa such as Eggerthella and Lachnospiraceae were similarly regulated in both formulas fed caesarian delivered infants and differed from breast fed infants. Ruminococcus 2, Erysipelotrichaceae, gut metagenome, Clade I and partly also Ruminococcaceae UCG 014, were enriched in vaginally delivered standard formula fed infants at T2. This supports the difference in alpha diversity observed in vaginally delivered babies fed a standard diet.'

We also performed the analysis of metabolites showing different concentration trends (T0-T2) according to delivery-diet groups accounting for fixed effects of diet and delivery and random effects of subjects, thus taking into account the individual baseline of each subject, using a linear mixed model. In Extended data Fig. 5, 20 statistically significant different metabolites in at least one group (by ANOVA p val<0.05) are reported for each individual infant over the course of the time (T0-T2).

Interestingly, we observed that most of these metabolites that characterized the different groups were also clusterizing with particular sets of microorganisms in Fig. 5.

This is reported on pages 13-14 lines 311-339 and reads as follows :

'We then analyzed metabolites showing different concentration trends (T0-T2) according to delivery-diet groups accounting for fixed effects of diet and delivery and random effects of subjects, thus taking into account the individual baseline of each subject, using a linear mixed model. In Extended data Fig. 5, we report 20 statistically significant different metabolites (by ANOVA among the diet delivery groups, at least one different from the others, $P < 0.05$) for each individual infant over the course of the time (T0-T2). As shown in Extended data Fig. 5, even taking into account baseline levels for individual subjects, metabolites in standard formula fed infants clearly demonstrated a trend which was different from that of the reference group, independent on the type of delivery. By contrast, trends of metabolites associated with the fermented formula fed group had values which were in some cases more similar to those of standard formula and in others more similar to the reference group. In particular, amino acids such as Valine and Threonine followed a trend similar in caesarean delivered standard formula fed infants and fermented formula fed infants, but in the latter group they were less abundant and significantly different ($P < 0.05$). Similarly, 2 keitoisocaproic acid, threose, 5 Hydroxyindoleacetic acid and Serine also followed a trend similar in standard formula fed infants and in fermented formula fed infants. By contrast, metabolites such as glyceril glycoside, glyceric acid, galactose and arabinose, displayed trends in ceasarean delivered fermented formula fed infants more similar to those of the reference groups. Interestingly, especially in formula fed infants the caesarian delivered babies had higher modifications of metabolite abundance over time. We also found some metabolites which were reduced or absent in formula fed infants as compared to the reference group such as aspartic acid, methionine, arabitol, mannobiose, 2-3 dihydroxy 2 methylpropanoic acid, propylene glycole, pentanedoic acid, stearic acid, galactofuranose and ribose. These metabolites were found significant primarily in the vaginal delivery reference group confirming their separation as a group versus all the other groups. When taking into account also the baseline for each individual infant compared to the analysis not corrected for baseline, some metabolites changed for statistical significance. This is somehow expected as breastfed infants baseline was different from that of formula fed infants as they had already breastfed for some days. This on one hand highlights which metabolites were modified over time, but on the other excludes those metabolites that are not modified over time due to continuation of the same dietary regimen. Hence, both analyses should be considered to infer conclusions.'

- Authors stated that they corrected their language throughout (in response to my comment that 'reduce' and 'increase' are not appropriate when referring to breastmilk, which is the biological norm). This is still misused throughout.

We actually went through the manuscript again to search for places where we have reported increase and reduced when referred to breast milk. We realized that it there were some left. We hope we have now corrected all of them.

In page 8 lines 212-214 the sentence:

*However, we observed a trend towards a reduced variability in breastfed infants at visit 2 **as compared to T0**, irrespective of the mode of delivery.*

Reduced refers to the same group at T2, thus we added 'as compared to T0'.

Here are some examples where we have modified the text, this is the new text

On page 7 lines 182-183

'formula fed infants, irrespective of the mode of delivery, showed reduced values of sIgA compared to the reference group'

On page 9 lines 245-249

'Clostridium innocuum group was enriched in formula fed infants irrespective of the type of formula or mode of delivery (p=0,031991571) while Veillonella was reduced as compared to breastfed infants microbiota irrespective of the mode of delivery (p=0,0091) as measured by post-hoc Tukey HSD analysis.'

On page 16 lines 405-407

'Indeed, we confirmed a previous report that standard formula fed infants had increased diversity of the microbiota ¹⁵, while the fermented formula behaved more similar to the reference group.'

On page 17 lines 442-444

'For example, we found that fermented formula fed infants had levels of myristic acid similar to those of breastfed infants, but reduced compared to standard formula fed infants.'

- In the introduction, the section on prebiotics/oligosaccharides is still unclear. Prebiotics can include GOS/FOS or synthetic HMOs. This is not clearly described.

We have further clarified this point. As the addition of GOS/FOS or synthetic HMOs was not the major focus of the manuscript we had only quoted this possibility without getting too much into details.

Anyway the reviewer will find the new description in the introduction on page 4 lines 98-106 and it is described as follows:

'The human milk is rich in oligosaccharides ^{10,11} which shape the infant microbiome ¹⁴. Thus often infant formulas are supplemented with prebiotics such as synthetic oligosaccharides like galacto-oligosaccharides, fructo-oligosaccharides and polydextrose ²⁴ that have a different structure from those of human milk and cannot completely mimic the anti-inflammatory activity of human milk-derived oligosaccharides in vitro ²⁵. They favor the development of the gut microbiota, and in particular of Bifidobacteria but what is the effect on the immune system has not been analyzed in vivo ²⁶. Synthetic oligosaccharides structurally identical to those of human milk have recently become available on the market showing prebiotic and anti-infective properties when added to formulas ²⁷.'

- Authors state they have updated the literature review but they have not highlighted the new references.

These are now highlighted as Yellow/underlined, those of the previous version only underlined

- I commented that "Some of the results described in the text are not supported by figures and conclusions throughout are very superficial." – The author response does not address this.

We revisited the whole discussion to make sure that there were no data not supported by figures.

Regarding the statement that the conclusions are superficial, we have gone through the literature to more clearly find possible implications of our findings. The reviewer has to however admit that very little is still known both on the microbiota and the metabolome to clearly infer conclusions from the data. Hence, the reviewer should now appreciate our effort to try to correlate our findings with the little evidence from the literature.

We have added a sentence on the possible role of myristic acid which is found enriched in standard formula fed infants but has amounts more similar to breastfed infants in standard formula:

These sentences have been introduced in the discussion:

On page 17 lines 442-449:

'For example, we found that fermented formula fed infants had levels of myristic acid similar to those of breastfed infants, but reduced compared to standard formula fed infants. Myristic acid has been associated with the appetitive response in newborns³⁸. However, we did not observe any change in body growth in the three different groups suggesting that its presence does not seem to influence food intake, at least in this time frame. Interestingly, intraperitoneal injection of myristic acid in a concentration similar to that found in the amniotic fluid to rats has been shown to induce an anxiolytic-like behaviour³⁹. Reduction of myristic acid in the fermented formula may give a health benefit to the newborn. '

We have discussed more on the microbes found to be enriched in formula fed infant.

And page 17 lines 450-462:

'In particular, we observed a series of metabolites enriched in standard formula fed infants that were correlating with Clostridium innocuum group, Erysipelotrichaceae, Eggerthella, Lachnospiraceae and Ruminococcus 2 which were enriched in formula fed infants. Unfortunately, little is known on most of these bacteria and it is hard to infer possible advantages or disadvantages of having an enrichment of these bacteria. Ruminococci for instance have been shown to produce short chain fatty acids such as propionate and butyrate⁴⁰ that have several beneficial effects both on epithelial cell proliferation and barrier functions⁴¹, however there are no specific reports on Ruminococcus 2. Similarly, regarding Clostridium innocuum there is only one report showing that it is associated with colobronchial fistula formation in Crohn's disease together with other potential pathogens⁴². Erysipelotrichaceae also have been associated with inflammatory states of the gut⁴³, but again individual species might have different activities. More indepth analysis should be carried out to deepen our knowledge on the enrichment of these genera in standard formula fed infants.'

We have inversely correlated the presence of Bacteroides, parabacteroides and Veillonella in the reference group with several metabolites enriched in standard formula fed infants. We have discussed more in details both the genera and the metabolites.

The reviewer can find this on pages 17-18 lines 465-481:

'Keitoisocaproic acid is a metabolic intermediate of L-leucine while Leucine and Valine are branched aminoacids and are the bases for the production of other essential amino acids such as glutamate, glutamine, aspartate⁴⁴. It is not clear why in the formula fed group there should be increase of these aminoacids as they are highly represented in milk proteins⁴⁴. One possibility is that Bacteroides, Parabacteroides or Veillonella might use up these aminoacids or transform them thus reducing their concentration and hence since these taxa are not enriched in formula fed infants these aminoacids accumulate. Notably, Bacteroides make up a substantial portion of the human gut microbiota and are considered necessary to refine the gut environment and transforminig it into one more hospitable for themselves and the other microrganisms⁴⁵. They reduce for instance oxygen levels intracellularly reducing at the same time both inflammatory reactive oxygen species and generating a strictly anaerobic environment that favors the growth of anaerobic microbes⁴⁶. In addition, Bacteroides are important for the digestion of complex molecules, including plant- or host-derived polysaccharides⁴⁵. As mentioned above, the human breast milk is rich in complex carbohydrates that are indigestible by our enzymes and this may explain the enrichment of Bacteroides in the reference group^{21,22}. Veillonella, on the other hand, uses lactate as its only source of carbon⁴⁷ and this may explain why lactose and lactulose are not abundant in the reference group.

- The new taxa barplot is completely unhelpful for assessing Bifidobacteria because a gradient colour scheme is used and I cannot tell Bifidobacteria apart from Bacteroides.

We have eliminated this barplot and substituted it with an analysis of the individual trends of Bifidobacteria at T0 and T2

Reviewers' comments:

Reviewer #4 (Remarks to the Author):

Table 3 – why are the values for the breastfed infants presented as a range rather than median \pm standard deviation?

Lines 216-218 – move this text to the discussion

Lines 602-614 Additional information is needed on the immunological analysis of the IgA and defensins. The results are expressed as ng/g or ug/g. Does the g refer to the starting material of the fecal sample? Was there a dilution factor used?

Also the values seem quite low, particularly for the alpha-defensins. Were all the linear range of the assays?

Typographical errors:

There are a number of grammatical issues throughout the manuscript.

Spelling of caesarean is inconsistent in the manuscript

Line 156, 157, 181, 209, 211, 212, 257, 270, 277, 286, 290, 303, 416, 420, 425, 430, 431, 530, 571, 573, 743, 747, 757, 761, Figure 2A and 2B, Figure 3B, Figure 4A and 4C, and Table 2 – enrolment should be enrollment

Line 201 – reduce should be reduces

We thank reviewer 4 for helpful critics. All changes are highlighted in yellow
In response to reviewer #4:

Table 3: – why are the values for the breastfed infants presented as a range rather than median \pm standard deviation?

Table 4 was modified according to this criticism. All data are now presented as median and range for easier comparison between groups. In the Statistical analysis section we added “or median and range” (line 797).

Lines 216-218 – move this text to the discussion

These sentences have been moved to the discussion (new lines:420-424)

Lines 602-614 Additional information is needed on the immunological analysis of the IgA and defensins. The results are expressed as ng/g or ug/g. Does the g refer to the starting material of the fecal sample? Was there a dilution factor used?

Also the values seem quite low, particularly for the alpha-defensins. Were all the linear range of the assays?

The g refers to the starting material of fecal samples collected by the study subjects. The methods section has been modified to better explain these procedures (see New lines 602-619):

For secretory immunoglobulin A (sIgA) and beta-defensin 2 (HBD-2) one g of fecal sample was diluted 1:1 (w/v) with PBS buffer (130 mM NaCl and 10mM sodium phosphate-buffered saline, pH 7.4). For alpha-defensin (HNP 1-3): one gr of fecal sample was diluted 1:0.5 (w/v) with the same buffer. This was necessary considering the detection limit of the kit that was commercially available when the investigation was performed (i.e.: 0.05 ng/g). All samples were then centrifuged at 13000 rpm for 15 minutes in 1.5 ml tubes. The supernatant was collected for quantification by ELISA, without further dilution. HNP 1-3 was measured by ELISA using a specific human kit (Hycult biotechnology, Uden, The Netherlands); HBD-2 by ELISA using a specific human kit (Phoenix Pharmaceuticals, Inc., Burlingame, CA, USA)(detection limit: 0.01 ng/g); and sIgA by indirect enzyme immunoassay for human samples(Salimetrics LLC, Carlsbad, CA, USA) (detection limit:2.5 μ g/g).

For LL-37 measurement, the sample (1 gr of fecal sample) was extracted with 60% acetonitrile in 1% aqueous trifluoroacetic acid (TFA) and then extracted overnight at 4°C. The extract was centrifuged and the supernatant stored at -20°C. LL-37 level was then measured, without dilution, by a commercially available specific ELISA kit for human samples (Hycult biotechnology, Uden, The Netherlands)(detection limit: 0.1 ng/g).

Typographical errors:

There are a number of grammatical issues throughout the manuscript.

The text was revised according to this criticism.

Spelling of caesarean is inconsistent in the manuscript

The text was revised according to this criticism.

Line 156, 157, 181, 209, 211, 212, 257, 270, 277, 286, 290, 303, 416, 420, 425, 430, 431, 530, 571, 573, 743, 747, 757, 761, Figure 2A and 2B, Figure 3B, Figure 4A and 4C, and Table 2 – enrolment should be enrollment

This actually depends whether is American or British English. We left it as enrolment. The editor will tell us what is the rule for the journal

Line 201 – reduce should be reduces

Corrected, thanks

REVIEWERS' COMMENTS:

Reviewers' comments:

Reviewer #4 (Remarks to the Author):

Table 3 – why are the values for the breastfed infants presented as a range rather than median \pm standard deviation?

Lines 216-218 – move this text to the discussion

Lines 602-614 Additional information is needed on the immunological analysis of the IgA and defensins. The results are expressed as ng/g or ug/g. Does the g refer to the starting material of the fecal sample? Was there a dilution factor used?

Also the values seem quite low, particularly for the alpha-defensins. Were all the linear range of the assays?

Typographical errors:

There are a number of grammatical issues throughout the manuscript.

Spelling of caesarean is inconsistent in the manuscript

Line 156, 157, 181, 209, 211, 212, 257, 270, 277, 286, 290, 303, 416, 420, 425, 430, 431, 530, 571, 573, 743, 747, 757, 761, Figure 2A and 2B, Figure 3B, Figure 4A and 4C, and Table 2 – enrolment should be enrollment

Line 201 – reduce should be reduces

We thank reviewer 4 for helpful critics. All changes are highlighted in yellow in response to reviewer #4:

Table 3: – why are the values for the breastfed infants presented as a range rather than median ± standard deviation?

Table 3 was modified according to this criticism. All data are now presented as median and range for easier comparison between groups. In the Statistical analysis section we added “or median and range” (line 605).

Lines 216-218 – move this text to the discussion

These sentences have been moved to the discussion (new lines: 315-319)

Lines 602-614 Additional information is needed on the immunological analysis of the IgA and defensins. The results are expressed as ng/g or ug/g. Does the g refer to the starting material of the fecal sample? Was there a dilution factor used?

Also the values seem quite low, particularly for the alpha-defensins. Were all the linear range of the assays?

The g refers to the starting material of fecal samples collected by the study subjects. The methods section has been modified to better explain these procedures (see new lines 497-511):

For secretory immunoglobulin A (sIgA) and beta-defensin 2 (HBD-2) one gram of fecal sample was diluted 1:1 (w/v) with PBS buffer (130 mM NaCl and 10mM sodium phosphate-buffered saline, pH 7.4). For alpha-defensin (HNP 1-3): one gram of fecal sample was diluted 1:0.5 (w/v) with the same buffer. This was necessary considering the detection limit of the kit that was commercially available when the investigation was performed (i.e.: 0.05 ng/g). All samples were then centrifuged at 13000 rpm for 15 minutes in 1.5 ml tubes. The supernatant was collected for quantification by ELISA, without further dilution. HNP 1-3 was measured by ELISA using a specific human kit (Hycult biotechnology, Uden, The Netherlands); HBD-2 by ELISA using a specific human kit (Phoenix Pharmaceuticals, Inc., Burlingame, CA, USA)(detection limit: 0.01 ng/g); and sIgA by indirect enzyme immunoassay for human samples (Salimetrics LLC, Carlsbad, CA, USA) (detection limit: 2.5mg/g). For LL-37 measurement, the sample (1 gram of fecal sample) was extracted with 60% acetonitrile in 1% aqueous trifluoroacetic acid (TFA) and then extracted overnight at 4°C. The extract was centrifuged and the supernatant stored at -20°C. LL-37 level was then measured, without dilution, by a commercially available specific ELISA kit for human samples (Hycult biotechnology, Uden, The Netherlands)(detection limit: 0.1 ng/g).

Typographical errors:

There are a number of grammatical issues throughout the manuscript.

The text was revised according to this criticism.

Spelling of caesarean is inconsistent in the manuscript

The text was revised according to this criticism.

Line 156, 157, 181, 209, 211, 212, 257, 270, 277, 286, 290, 303, 416, 420, 425, 430, 431, 530, 571, 573, 743, 747, 757, 761, Figure 2A and 2B, Figure 3B, Figure 4A and 4C, and Table 2 – enrolment should be enrollment

This actually depends whether is American or British English. We left it as enrolment. The editor will tell us what is the rule for the journal.

Line 201 – reduce should be reduces

Corrected, thanks.